# Potential $CO_2$ measurement capabilities of a transportable Near Infrared Laser Heterodyne Radiometer (LHR)

Marie Thérèse El Kattar[1], Tingting Wei[1], Aditya Saxena[1], Hervé Herbin[2], and Weidong Chen[1]

[1]Laboratoire de Physico-Chimie de l'Atmosphère (LPCA), Université du Littoral Côte d'Opale, 189A, Avenue Maurice Schumann, 59140, Dunkerque

[2]UMR 8518 – LOA – Laboratoire d'Optique Atmosphérique, Univ. Lille, CNRS, F-59000, Lille, France

*Correspondence to*: Marie Thérèse El Kattar (marie-therese.el-kattar@univ-littoral.fr), Hervé Herbin (herve.herbin@univ-lille.fr) and Weidong Chen (chen@univ-littoral.fr)

**Abstract.** In this study, heterodyne detection enables high spectral resolution, which in turn enhances the vertical sensitivity of ground-based $CO_2$ measurements. The system's compact and portable design makes it particularly well-suited for deployment in field campaigns. An all-fiber coupled laser heterodyne radiometer (LHR), using a wideband tunable external cavity diode laser (1520-1580 nm) as local oscillator laser was developed for $CO_2$ measurements. Optimal absorption lines and transmission spectra of the LHR was achieved by using a balanced photodetector to suppress the relative intensity noise of the local oscillator laser. This work aims to quantify how the LHR contributes to measuring tropospheric $CO_2$ abundances in the atmospheric column from the ground. Here, we demonstrate the LHR's ability to measure $CO_2$ vertical profiles through an extensive analysis of information content, channel selection, and error budget estimation. This comprehensive analysis relies on the radiative transfer model ARAHMIS, developed at the Laboratoire d'Optique Atmosphérique (LOA). Additionally, we present a comparison of the LHR with other ground-based instruments, such as the EM27/SUN and the IFS125HR from the TCCON network. Furthermore, this work supports ongoing MAGIC (Monitoring of Atmospheric composition and Greenhouse gases through multi-Instruments Campaigns) campaigns focused on greenhouse gas monitoring and the validation of current and future space missions such as MicroCarb and CO2M.

## 1 Introduction

Developing robust and affordable techniques for the accurate measurement of greenhouse gas (GHG) concentrations is essential for monitoring their spatiotemporal variability and supporting the study of emission sources, sinks, and atmospheric transport processes. Alongside spaceborne instruments such as OCO-2 (Eldering et al., 2017), which offer global coverage and high GHG column abundance accuracy, there's a growing need for compact, portable, and cost-effective instruments that can validate satellite observations but also monitor major GHGs in the atmospheric column. In addition to compactness, high mobility and low cost, these devices must have extremely high spectral resolution to meet GHG observation requirements (IPCC 2023, AR6 WGI, Ch.1 & Ch. 10). The Bruker IFS125HR Fourier Transform Spectrometer (FTS), with a spectral resolution of approximately 0.02 cm⁻¹, is the main instrument used by the Total Carbon Column Observing Network (TCCON)

(Wunch et al., 2010). However, this spectrometer's limitations for field campaigns hinder its broader use in ground-based atmospheric measurements worldwide. The COllaborative Carbon Column Observing Network (COCCON) complements TCCON by deploying portable Fourier-transform spectrometers, specifically Bruker's EM27/SUN instruments. These spectrometers are relatively easy to operate and enable measurements in locations inaccessible to larger systems, with a spectral resolution of 0.5 cm$^{-1}$ (Table 3), a trade-off from their compact design which limits the maximum optical path difference. While their portability allows for flexible deployment, maintaining network-wide consistency and coordination remains a significant logistical and technical achievement (Frey et al., 2019). Moreover, several studies have directly compared the performance of the high-resolution IFS125HR with the portable EM27/SUN spectrometers, including Pak et al. (2023) and Herkommer et al. (2024), showing that $CO_2$ retrievals from the EM27/SUN differ by only about 0.1%, a remarkable result considering its lower spectral resolution. In contrast, heterodyne detection offers a cost-effective, highly mobile system that enhances vertical sensitivity limits and achieves exceptional spectral resolution (Weidmann, 2021). While the overall system cost depends significantly on the choice of laser and detector, the prototype LHR developed in this study is approximately 20% of the cost of an EM27/SUN, making it a promising complementary tool for targeted ground-based observations. This suggests that heterodyne spectro-radiometers could serve as a valuable addition not only to TCCON's measurements (Palmer et al., 2019), but also for the EM27/SUN based COCCON network. While commercially heterodyne spectroradiometers are currently unavailable, scientific groups worldwide are presenting their achievements in the development and application of these instruments in the near-infrared (NIR) spectral range (Zenevich et al., 2020). An all-fiber coupled laser heterodyne radiometer (LHR) has been developed at the Laboratoire de PhysicoChimie de l'Atmosphère (LPCA) for measuring carbon dioxide ($CO_2$) and water vapor ($H_2O$) concentrations in the atmospheric column (Wang et al. 2023). The LHR uses a broadband tunable external cavity diode laser operating between 1520-1580 nm as a local oscillator (LO) laser. To improve signal to noise ratio in LHR spectra, a balanced photodetector is employed to suppress the laser relative intensity noise (RIN) of the LO laser.

This study presents the principle of the LHR experimental setup and quantifies its potential for $CO_2$ retrieval. The structure of the paper is as follows: Section 2 describes the setup and technical characteristics of the instrument, Section 3 provides a detailed explanation of the forward model, state vector, and a complete error analysis. We present in Section 4, a comparison with the other FTS instruments for the retrieval of $CO_2$ building on earlier research (El Kattar, Auriol, and Herbin 2020). Section 7 presents the channel selection methodology employed in this work, which is essential for determining the most suitable channels for measurement. The study concludes with a summary of the results and explores future directions, such as improving measurement precision, and particularly the $CO_2$ retrieval.

## 2 Experimental setup

The LHR used in the present work, depicted in **Figure 1**, is designed to measure atmospheric $CO_2$ and $H_2O$ concentrations by measuring their absorptions of the sunlight in the NIR. Solar radiation is captured using a portable solar tracker (STR-21 G; EKO Instruments Co., Ltd.), which continuously tracks the sun's position. A mechanical chopper (MC2000B; Thorlabs, Inc.)

modulates the sunlight to enable phase-sensitive detection via the lock-in amplifier, isolating the heterodyne signal from low-frequency noise. This modulated sunlight is combined with light from a tunable external cavity diode laser (TUNICS-BT 3642 HE CL; NetTest), which serves as the LO. The laser operates at room temperature, with a tunable wavelength range of 1520–1580 nm and a maximum power of 5 mW. A fiber collimator (F810APC-1550, Thorlabs, Inc.) collects sunlight into a 2-meter single-mode fiber (SMF-28-J9, Thorlabs, Inc.) and is mounted on the solar tracker with a numerical aperture of 0.24. On sunny days, the solar power collected in the single-mode fiber can reach 7.9 µW. The modulated radiation is split by a single-mode fiber splitter with a 40:60 beam splitting ratio. The 40% power is measured with a photodetector (PDA20CS-EC; Thorlabs, Inc.) to monitor sunlight intensity variation during measurements. The 60% power is mixed with the LO laser for heterodyne detection. In addition, a balanced amplified photodetector (PDB425C; Thorlab Inc.) is used to reduce laser RIN resulting from the LO laser. For this purpose, the LO laser output light is split into two beams with a 50:50 fiber splitter, 50% used to mix the sunlight and the other 50% used for balanced detection. The beating signal at radio frequency (RF) from the balanced photodetector passes through a band-pass filter with an effective bandwidth of 24–95 MHz. Subsequently, a Schottky diode (a square-law detector), is used to extract the absorption signature, which corresponds to the envelope of the RF beat signal. This type of detector produces an output proportional to the square of the input signal's amplitude. The electronic bandwidth of the detector is $B_{IF} = 100$ kHz to 2 GHz, enabling effective heterodyne detection. The resulting output signal is then demodulated using a lock-in amplifier (LIA-MV-150; FEMTO Inc.). A data acquisition card (DAQ) (USB-6366; NI Inc.) digitizes the spectral signal that is then transferred to a laptop for further data processing and retrieval.

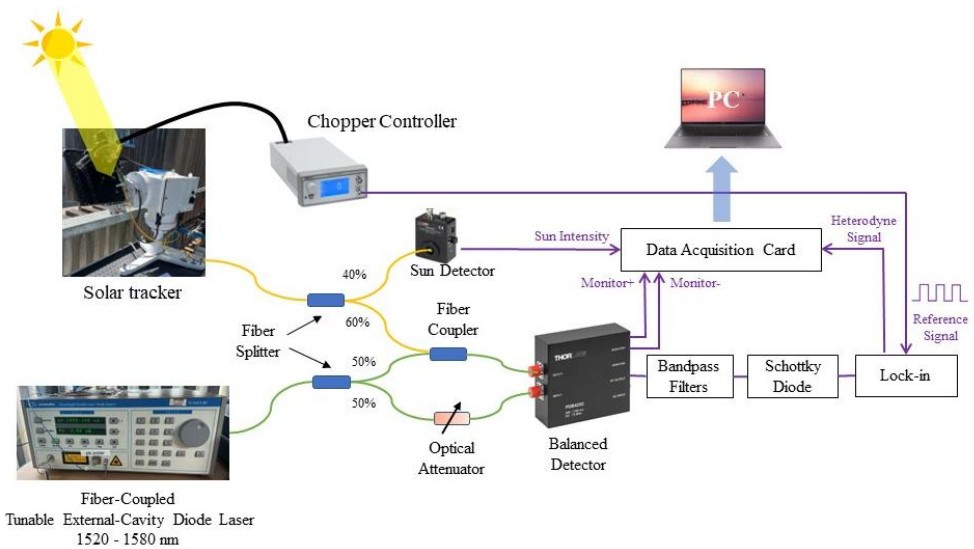

Figure 1: Schematic of the developed LHR. Each percentage represents the proportion of radiation beams split or combined.

To achieve accurate measurements of relative transmittance using the LHR, it is essential to extract spectral signals with a high Signal-to-Noise Ratio (SNR). This LHR system is designed to operate in the shot noise–limited regime by optimizing the local oscillator laser power, such that the total system noise is dominated by LO-induced shot noise (Sun et al., 2024). Therefore, the thermal noise can be ignored. Meanwhile, a balanced detector in LHR is used for heterodyne signal detection, eliminating the relative intensity noise of the local oscillator laser. A full discussion of the noise model goes beyond the scope of the present manuscript and will be addressed in a forthcoming technical paper.

The SNR for the measurement via coherent detection can be written as follows:

$$SNR = \frac{2T_0\eta\sqrt{\Delta f \tau}}{2\eta + \exp(h\upsilon/kT_S) - 1}, \tag{1}$$

where $\tau$ is the integration time, $\Delta f$ is the filter bandwidth, $T_0$ is the transmission efficiency, $\eta$ is the quantum efficiency of the photodetector, $T_S$ is the temperature of the heat source, $k$ is the Boltzmann constant, $\upsilon$ is the wavenumber and $h$ is the Planck constant. For a typical sunlight measurement, with an integration time of 100 ms, $\Delta f$ of 52 MHz, $T_0$ of 1, a quantum efficiency of 0.81 (provided by the manufacturer) and $T_S$ of 6011 K on average, we find an average theoretical SNR of 710 for the spectral domain covered by the LHR. The absorption spectra obtained from these measurements are illustrated in **Figure 2**. The actual measured SNR is approximately 200, based on a single scan, in contrast to the FTIR measurements where multiple scans are averaged. The reduced SNR can be attributed to several factors, primarily the absence of spectral averaging. Additional contributors include suboptimal detector performance such as lower-than-expected quantum efficiency, elevated dark current, and electronic noise sources including amplifier and digitizer interference. While current measurements yield a lower SNR, an SNR of 710 is achievable through additional scan averaging or improved detector performance. We therefore use SNR = 710 to assess the theoretical information content under optimal conditions, which will be targeted in future measurement campaigns.

## 3 Theory

In order to determine and evaluate the capacity and the performance of the developed LHR, an information content study (IC) is conducted to assess its potential for GHG retrieval and compare it with other well-established techniques for worldwide observation.

### 3.1 The forward model

To accurately simulate the transmittances observed by the LHR, the line-by-line radiative transfer algorithm ARAHMIS was used across a broadband NIR spectrum of 1.567–1.577 µm. The absorption spectrum of gases is derived using the updated HITRAN 2020 database (Gordon et al., 2022) , with spectral lines represented by Voigt profiles. The resulting spectrum is convolved with a Gaussian Instrument Line Shape (ILS), which reflects the optical and detection characteristics of the LHR

system. In addition, absorption continua for water vapor ($H_2O$) and carbon dioxide ($CO_2$) are incorporated using the MT-CKD model (Clough et al., 2005). The incident solar spectrum is derived from the pseudo-transmittance spectra for direct sunlight originating from the center of the solar disk, as provided by Toon (2015), and subsequently interpolated onto the LHR's spectral range. The Planck function is calculated across the LHR spectral domain using a custom routine developed at LATMOS (Meftah et al., 2018), to account for the significant variation in effective brightness temperature with wavenumber. This routine is based on the SOLAR-ISS spectrum, a high-resolution solar reference spectrum constructed by combining existing solar datasets with SOLAR/SOLSPEC measurements, using well-characterized slit functions. SOLAR-ISS provides an accurate representation of solar irradiance during the 2008 solar minimum, particularly across the ultraviolet, visible, and infrared regions. Accurate determination of the spectrometer's line-of-sight (LOS) is crucial for determining the spectral absorption of solar radiation as it propagates through the atmosphere during the retrieval process of gases. To achieve this, the timing and duration of each measurement are recorded, allowing the calculation of the Solar Zenith Angle (SZA) using the methodology described in Michalsky (1988).

Measurements are conducted in Dunkirk (51.035°N, 2.369°E) under clear sky conditions in August 2022. The calculations depend on the concentration of the target atmospheric profile, along with associated data profile such as temperature, pressure, and relative humidity, which are obtained from a nearby PTU300 Vaisala radiosonde, with manufacturer-specified uncertainties of ±0.2°C for temperature, ±0.3 hPa for pressure, and ±1% for relative humidity. A priori profiles of $CO_2$ and $H_2O$ used to construct the state vector and prior covariance matrix are derived respectively from the AirCore launches from the MAGIC campaigns (see Section 4.1) and the Orléans TCCON station, which is the closest operational site to Dunkirk.

**Figure 2** displays the results of ARAHMIS's simulation compared to a typical measurement of the mid-infrared band by the LHR. Additionally, the impact of the solar spectrum, $CO_2$, and $H_2O$ is shown, demonstrating the strong consistency between the forward model simulation and the LHR measurements under clear-sky conditions.

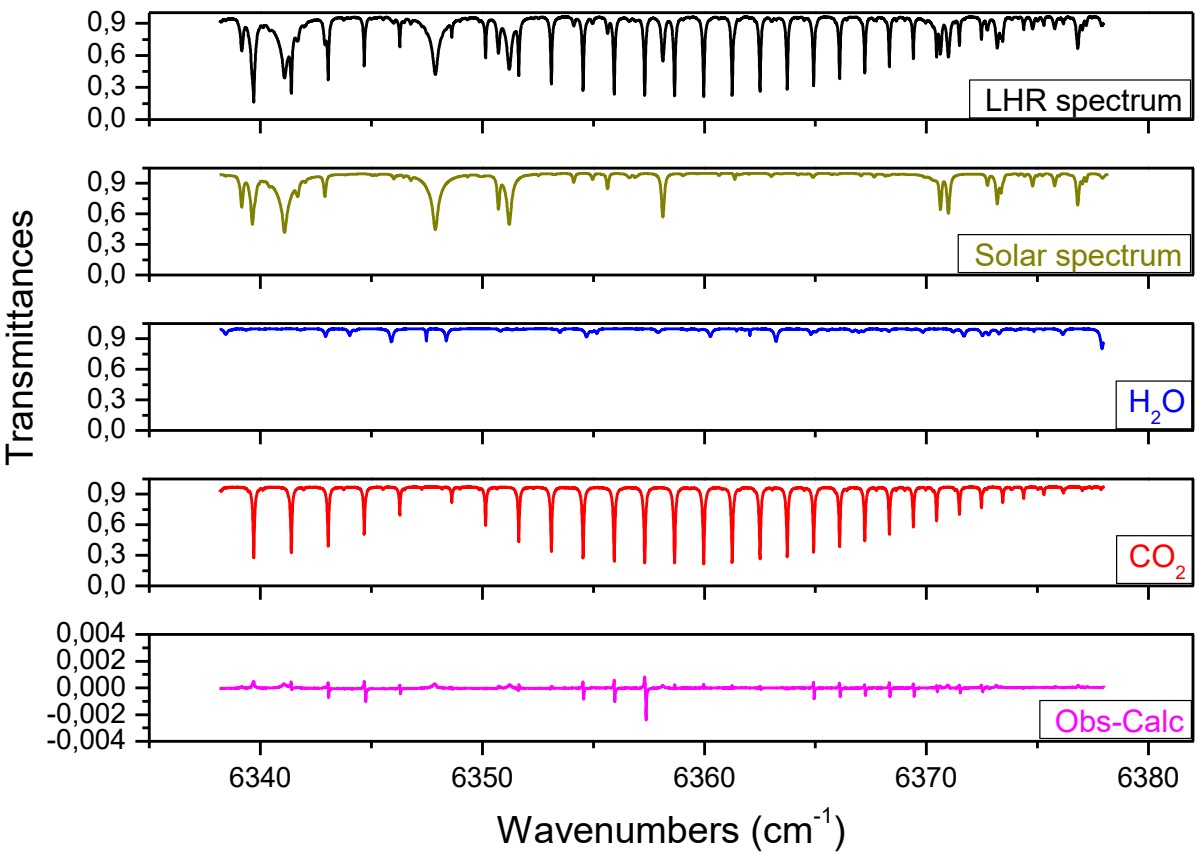

Figure 2: Comparison of measured and simulated LHR transmittance spectra under clear-sky conditions in Dunkirk, for an SZA of 55° and a total integration time of 15 minutes. The measured spectrum is shown in black, while the simulated spectra , computed with the line-by-line forward model ARAHMIS, are shown in blue for $H_2O$ and in red for $CO_2$. The simulations incorporate the solar pseudo-transmittance spectra from Toon 2015in gold. The residuals between the measurement and simulation are plotted in magenta.

### 3.2 Theoretical basis of Information Content (IC)

Following the computation of the forward model, we apply the framework developed by Rodgers (2000), which incorporates the optimal estimation theory employed in the retrieval process. This theory has been extensively discussed in prior works (Herbin et al., 2013) and briefly summarized here. As this study builds on previous research, certain sections are condensed, focusing only on essential details. For a comprehensive explanation, please refer to El Kattar, Auriol, and Herbin (2020). In this study, the state vector $x$ consists primarily of the vertical profile of $CO_2$ volume mixing ratios (VMR) on a fixed altitude grid extending from the surface to 40 km at 1 km vertical resolution. Depending on the retrieval scenario, the state vector may also include additional parameters, such as a scaling factor for atmospheric temperature. The measurement vector $y$ comprises calibrated radiance spectra derived from observed solar absorption, computed by multiplying the solar spectrum

(transmittance) with the SOLAR-ISS spectrum (see Section 3.1). Prior to retrieval, all measured spectra are corrected for spectral shift and solar abscissa scale by calibrating against a stable, unsaturated $H_2O$ absorption line. A scaling factor $\alpha$ is derived from the observed and theoretical line positions to correct the solar spectral abscissa. This correction is performed during preprocessing and is not part of the state vector.

In cases where the atmosphere is divided into discrete layers, the forward radiative transfer equation establishes an analytical relationship between the observation set $y$ (radiance) and the true atmospheric parameter vector $x$ (vertical $CO_2$ concentration profiles to be retrieved):

$$y = F(x; b) + \varepsilon, \tag{2}$$

Here, $F$ is the forward radiative transfer function (ARAHMIS code), $b$ represents fixed parameters influencing the measurement (atmospheric temperature, interfering species, viewing angle), and $\varepsilon$ is the measurement error vector.

The Jacobian matrix $K$, also referred to as the weighting function, represents the partial derivatives of the $i$th spectral channel in the measured spectrum with respect to each ($j$) element of the state vector: $K_{ij} = (\partial F_i / \partial x_j)$.

The gain matrix $G$, is defined as the matrix whose rows correspond to the derivatives of the retrieved state with respect to the spectral points, as follows:

$$G = \partial \hat{x} / \partial y = (K^T S_\varepsilon^{-1} K + S_a^{-1})^{-1} K^T S_\varepsilon^{-1}, \tag{3}$$

where $S_a$ stands for the a priori covariance matrix, reflecting our knowledge of the state space prior to measurement, while $S_\varepsilon$ denotes the covariance matrix encompassing errors from both the measured signal and the forward model. The superscript T denotes matrix transposition.

The averaging kernel matrix $A$, which quantifies the sensitivity of the retrieved state to the true state, is given by:

$$A = \partial \hat{x} / \partial x = GK, \tag{4}$$

Each row of $A$ corresponds to one retrieved parameter and indicates how changes in the true state at various altitudes influence the retrieval. At any altitude, the peak of an averaging kernel row marks the altitude of the highest sensitivity, while its full width at half maximum (FWHM) estimates the vertical resolution. The Degrees Of Freedom (DOFs) of the signal, given by

the trace of matrix $A$, represents the number of independent pieces of information retrievable from observations. In an ideal retrieval with an optimal inverse method, the averaging kernel matrix $A$ would equal the identity matrix, and the DOFs would match the size of the state vector. Hence, each parameter to be retrieved corresponds to a partial degree of freedom, represented by the respective diagonal element of $A$.

The posterior error covariance matrix $S_x$, characterizes the state space post-measurement. This total retrieval error can be

decomposed into three distinct contributions (Rodgers 2000):

$$S_x = S_{smoothing} + S_{meas.} + S_{fwd.mod.}, \tag{5}$$

In the above equation, the smoothing error covariance matrix $S_{smoothing}$ captures the vertical sensitivity of the measurements to the retrieved profile, with $I$ being the unity matrix:

$$S_{smoothing} = (A - I)S_a(A - I)^T, \tag{6}$$

$S_{meas.}$ reflects the influence of the measurement error covariance matrix $S_m$, derived from spectral noise, on the posterior error covariance matrix $S_x$. $S_{meas}$ is calculated from the spectral noise as follows:

$$S_{meas.} = GS_mG^T, \tag{7}$$

Finally, $S_{fwd.mod.}$ represents the contribution to the posterior error covariance matrix via $S_f$ the forward model error covariance matrix, which accounts for uncertainties in non-retrieved model parameters:

$$S_{fwd.mod.} = GK_bS_b(GK_b)^T = GS_fG^T, \tag{8}$$

where $S_b$ represents the error covariance matrix of the non-retrieved parameters, and $K_b$ is the Jacobian with respect to the non-retrieved parameters. The two matrices, $A$ and $S_x$, together define the information content of the LHR retrieval.

## 4 Application

The IC analysis uses simulated radiance spectra from the current LHR. The initial $CO_2$ vertical concentrations in the state vector $x_a$ follow the criteria in Section 3.1, divided into 40 layers from ground level to 40 km at 1 km intervals. Non-retrieved parameters, such as water vapor profile, temperature, and SZA, are included as outlined in Section 4.3. A priori values and their variability are detailed in **Table 1** and discussed in subsequent sections.

### 4.1 A priori covariance matrix

The a priori error covariance matrix $S_a$ can be evaluated using in-situ data or climatology. We assume firstly that $S_a$ is a diagonal matrix that are common for space-based retrievals (De Wachter et al., 2017), with each diagonal element ($S_{a,ii}$) defined as:

$$S_{a,ii} = \sigma_{a,i}^2 \text{ with } \sigma_{a,i} = x_{a,i} \cdot \frac{p_{error}}{100}, \tag{9}$$

where $\sigma_{a,i}$ denotes the standard deviation in the Gaussian statistics framework, and the subscript $i$ corresponds to the $i$th parameter of the state vector, and $p_{error}$ is the profile a priori error. The $CO_2$ profile a priori error is derived aligns with prior studies using FTS instruments (El Kattar, Auriol, and Herbin 2020).

Nevertheless, the correlation between vertical layers is primarily reflected in the off-diagonal elements of the covariance matrix. For this reason, we also employ an a priori covariance matrix: the $H_2O$ covariance matrix is constructed using climatological data from the TCCON Orléans station for the period 2016–2023 (https://data.caltech.edu/records/gexfp-a3461),

while the $CO_2$ and temperature covariance matrices are derived from publicly available AirCore measurements collected over the same period during the MAGIC campaigns ([https://data.ipsl.fr/repository/MAGIC/](https://data.ipsl.fr/repository/MAGIC/)). The use of these two a priori covariance matrices in the LHR retrieval is presented in the following sections.

## 4.2 Measurement error covariance matrix

The measurement error covariance matrix is calculated based on instrument performance and accuracy, linked to the radiometric noise characterized by the SNR (discussed in Section 2). We assume that this matrix is diagonal, with the $i$th diagonal element computed as:

$$S_{m,ii} = \sigma_{m,i}^2 \text{ with } \sigma_{m,i} = \frac{y_i}{SNR}, \tag{10}$$

where $\sigma_{m,i}$ is the standard deviation of the $i$th measurement ($y_i$) in vector $y$, representing the noise equivalent spectral radiance. The LHR's theoretical SNR is estimated at ~710, with additional instrument details provided in **Table 3**.

## 4.3 Characterization and accuracy of non-retrieved parameters

Errors from non-retrieved parameters are complex, primarily arising from water vapor and temperature effects in our scenario (see **Figure 2**). We assume vertically uniform uncertainties for both. Notably, water vapor is treated as a non-retrieved parameter in this study.

We set the $H_2O$ column uncertainty ($p_{Cmol}$) at 10% instead of using a profile error. For temperature, we assumed a realistic uncertainty of $\delta T = 1 K$ for each layer, consistent with typical ECMWF assimilation values. The SZA uncertainty is set at 0.35°, reflecting typical solar angle variations during measurements. These values are summarized in **Table 1**.

The total forward model error covariance matrix ($S_f$), assumed diagonal, is the sum of contributions from each diagonal element, with the $i$th diagonal element ($S_{f,ii}$) expressed as:

$$S_{f,ii} = \sum_{j=1}^{n\ level} \sigma_{f,T_j,i}^2 + \sigma_{f,H_2O,i}^2 + \sigma_{f,SZA,i}^2, \tag{11}$$

This section excludes spectroscopic effects like line parameters, line mixing, and continuum errors, which are discussed in Section 3.4.2 in relation to the $X_G$ column estimation.

| State vector elements | $T$ | $H_2O$ | SZA | $CO_2$ |
|---|---|---|---|---|
| A priori profiles | AirCore launch 2022 | ERA5 reanalysis | 10/80° | AirCore launch 2022 |
| Diagonal a priori uncertainty ($p_{error}$) | 1 K per layer | 10% | 0.35° | 1.3%-8% |
| Non-diagonal a priori uncertainty | AirCore dataset 2016-2023 | ERA5 reanalysis | 0.35° | AirCore dataset 2016-2023 |

Table 1: State vector parameters where $H_2O$, T and $CO_2$ are profiles and the value of SZA is scalar.

## 5. Information Content and Uncertainty Estimation for the LHR

We perform an information content analysis on the $CO_2$ broadband spectrum. The state vector includes gas concentrations at each level from 0 to 40 km, matching FTS and MAGIC instrument altitudes (balloons exceeding 25 km). This setup estimates each gas profile individually, with other atmospheric parameters and gas profiles known from ancillary data with specific uncertainties. Two SZAs, 10° and 80°, are chosen to illustrate the impact of solar optical path on sensitivity (depends on viewing geometry). Detailed discussions on averaging kernels and error budgets follow in subsequent subsections.

### 5.1 Estimation of Averaging Kernels and error budget

The left panel of **Figure 3a** shows the averaging kernel $A$ and the right panel shows the total posterior error $S_x$ for $CO_2$ at a 10° SZA. Results for 80° SZA are omitted due to similar vertical distributions, though slight differences in amplitude exist. These variations are discussed to quantify the viewing geometry's impact. $A$, derived independently using Section 4 variability, reflects the partial degree of freedom at each level. Each colored line represents the row of A at each vertical grid layer. Each peak of A represents the partial degree of freedom of the gas at each level that indicates the proportion of the information provided by the measurement. Values near 1 indicate measurement-dominated information, while values near 0 suggest prior knowledge dominance. Averaging kernels are close to 1 in the first layer and remain significant between 1 and 10 km, indicating a meaningful improvement of the information, while approaching 0 above 15 km. The measurement offers insights into the $CO_2$ levels from the ground up to 20 km, but at higher altitudes, information primarily relies on prior knowledge due to reduced sensitivity of these gases in the upper atmosphere. This contrast is clear in the error analysis: the a posteriori total error (solid black line) is much smaller than the a priori error (red line) in the lower atmosphere (0-15 km), indicating an improved $CO_2$ profile knowledge. Above 15 km, however, the total posterior error equals the prior error, signaling reduced sensitivity at high altitudes. Additionally, the errors associated with measurement and the forward model's dependence on non-retrieved parameters are minimal compared to other errors, indicating negligible SNR error. Nevertheless, smoothing error outweighs other errors, particularly beyond 20 km, indicating strong reliance on the a priori profile at higher altitudes and minimal contribution from measurements.

To overcome this problem, we conducted a similar study using a non-diagonal a priori covariance matrix (see Section 4.1). This approach yields a more homogeneous vertical distribution across all layers (left panel, **Figure 3b**). While the overall shape of the error budget remains similar to that of the variance, both the a priori and a posteriori uncertainties are significantly reduced. The measurement and forward model errors remain somewhat weak (see **Table 2**). Notably, although the smoothing error is smaller, the increased constraint leads to a greater propagation of smoothing error across vertical layers. This trade-off results in a reduced total uncertainty but also leads to lower DOFs.

In **Table 2**, the DOFs for $CO_2$ are presented for both 10° and 80° angles. The table indicates that, with a diagonal prior covariance matrix, four to five partial tropospheric columns for $CO_2$ can be retrieved. As anticipated, the DOFs are slightly

higher at $80°$ due to the longer solar optical path through each layer. The total profile error, derived from the diagonal of $S_x$, is discussed in the next section. Overall, the LHR demonstrates high vertical sensitivity and reduced error in the lower

atmospheric layers, where satellite instruments typically have limited sensitivity. However, when employing a non-diagonal a priori covariance matrix, one to two less partial tropospheric column is retrieved, but the error budget estimation is significantly improved. This highlights the importance of using a climatological a priori covariance matrix to reduce the errors in retrieved partial columns.

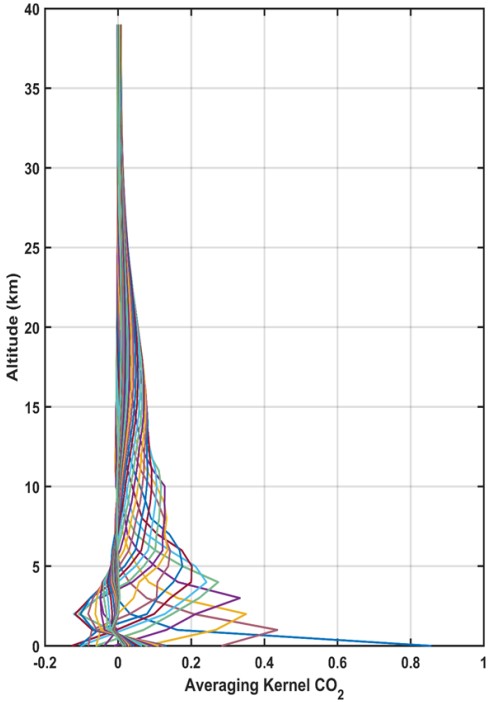
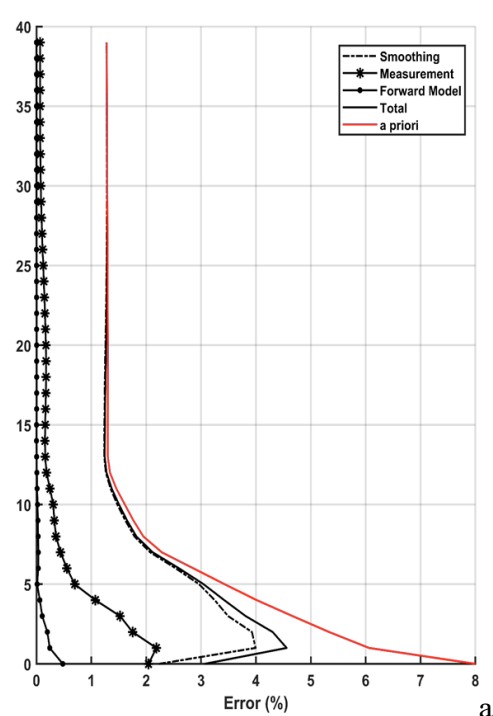

a)

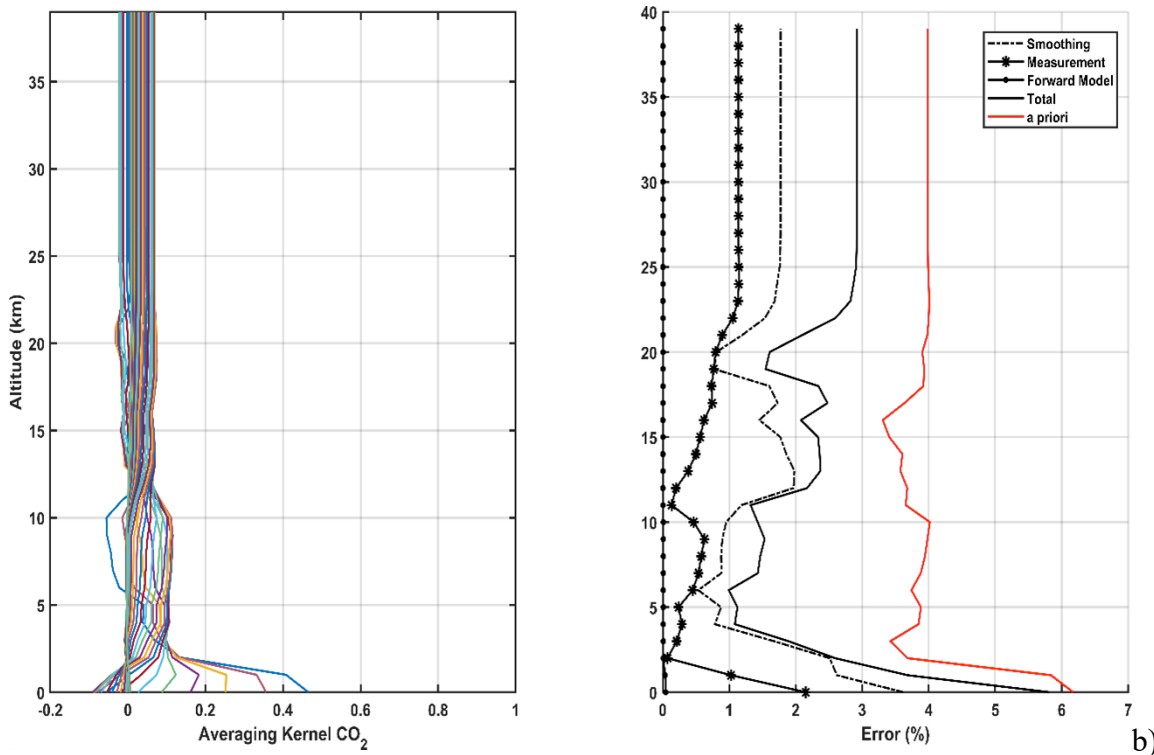

Figure 3: Averaging kernels and error budgets for $CO_2$ vertical profiles using the LHR for the a) diagonal a priori matrix and b) non-diagonal a priori matrix for a SZA of 10°. The red and solid black lines (in the right panels) stand for the prior $S_a$ and posterior $S_x$ errors respectively; the smoothing ($S_{smoothing}$), measurement ($S_{meas.}$) and forward model parameter ($S_{fwd.mod.}$) errors are dash-dotted, dash-starred and dotted, respectively.

**5.2 Estimation and uncertainty of integrated profiles**

Similar to the LHR, ground-based instruments such as the IFS125HR (TCCON) and EM27/SUN (COCCON) operate in the NIR and derive column-averaged dry-air mole fractions ($X_G$ for gas $G$) by observing simultaneously the $O_2$ columns. $X_G$ is computed as the ratio of the gas slant column to the $O_2$ slant column from the same spectrum. Since the LHR is narrow-banded and does not cover the absorption lines of $O_2$, a different method is needed to calculate this ratio. Following the NDACC

network (De Mazière et al., 2018), $X_G$ is calculated without using oxygen as a reference. Following the method outlined in Wunch et al., (2010) and used in Zhou et al., (2019), $X_G$ for $CO_2$ can be calculated as follows:

$$X_G = \frac{column_G}{column\ dry\ air}, \tag{12}$$

$$column\ dry\ air = \frac{P_S}{g_{air}m_{air}^{dry}} - column_{H_2O}\frac{m_{H_2O}}{m_{air}^{dry}}, \tag{13}$$

where $m_{H_2O}$ and $m_{air}^{dry}$ are the mean molecular masses of water and dry air, respectively. $P_S$ is the surface pressure and $g_{air}$

the column-averaged gravitational acceleration. Thus, $X_G$ can be calculated if all necessary parameters are available,

particularly in field measurements with data from balloons and radiosondes (e.g., temperature, relative humidity, surface pressure). It's important to note that TCCON's method removes systematic errors common to both the target gas and $O_2$ columns, which is not possible here. In our current LHR configuration, we are not yet able to retrieve $O_2$ columns, as we lack a laser source covering the 1.26 μm $O_2$ absorption band. Procuring such a laser is a planned future upgrade to enable direct 285 $XCO_2$ retrieval via the $CO_2/O_2$ column ratio, consistent with the approach used in TCCON and COCCON. In the absence of an $O_2$ measurement, we do not currently compute $XCO_2$, and the uncertainty budget is expressed in terms of vertically integrated $CO_2$ profile uncertainty, rather than in terms of $XCO_2$.

The integrated profile uncertainty is calculated by summing the concentration of each layer, weighted by the dry air column. **Table 2** displays the propagated uncertainties for both zenith angles, comparing results obtained with diagonal and non-290 diagonal a priori covariance matrices. At 10°, the total uncertainty decreases from 2.74% (diagonal) to 2.40% (non-diagonal), while at 80°, it reduces from 2.31% to 1.95%. The lower uncertainty at 80° is attributed to the longer atmospheric path length, which improves information distribution across layers. Breaking down the error contributions, smoothing error is the dominant source, accounting for 2.5% (diagonal) and 1.72% (non-diagonal) at 10°, and 1.91% (diagonal) and 1.49% (non-diagonal) at 80°. Measurement errors are smaller but still notable, decreasing from 0.99% to 0.66% at 10°, and from 1.05% to 0.44% at 295 80°. Errors due to non-retrieved parameters such as $H_2O$, temperature, and solar zenith angle are minimal when using the non-diagonal covariance matrix (0.015% at 10° and 0.017% at 80°) compared to the diagonal case (0.114% and 0.311%, respectively). The DOFs correspondingly decrease when using the non-diagonal matrix, from 4.13 to 2.79 at 10° and from 5.15 to 2.89 at 80°, reflecting a stronger constraint on the retrieval. It is important to note that spectroscopic uncertainty, which is systematic in nature, is not included here due to its complexity. This uncertainty varies across different absorption lines used 300 in the retrieval, with values listed in HITRAN.

| Error | $CO_2$ | |
|---|---|---|
| SZA | 10° | 80° |
| | diag/non-diag | diag/non-diag |
| Smoothing | 2.5/1.72 | 1.91/1.49 |
| Measurement | 0.99/0.66 | 1.05/0.44 |
| Non-retrieved parameters | 0.114/0.015 | 0.311/0.017 |
| Total | 2.74/2.4 | 2.31/1.95 |
| DOFs | 4.13/2.79 | 5.15/2.89 |

Table 2: The integrated profile errors and DOFs for the $CO_2$ profile for the LHR for the two SZAs and for the two covariance matrices. The uncertainties are expressed as percentages (%).

## 6 Comparison with existing networks

A similar previous study was performed for ground based Fourier Transform spectrometers, including the TCCON's IFS125HR, COCCON's EM27/SUN, and another EM27/SUN spectrometer operating in the middle infrared region called CHRIS (El Kattar, Auriol, and Herbin, 2020), as part of the MAGIC campaigns. Here, we compare these instruments with the LHR, but first we present in **Table 3** the different characteristics of the various instruments involved in this study.

|  | Resolution(cm$^{-1}$) | OPD (cm) | $CO_2$ micro-window (cm$^{-1}$) | SNR/integration time |
|---|---|---|---|---|
| LHR | 0.0047 | Fiber-coupled | 6338-6378 | 710/15 mins |
| EM27/SUN | 0.5 | 1.8 | 6173-6390 | 1080/1 min |
| IFS125HR (TCCON) | 0.02 | 45 | 6300 band | ~750/~3 mins |
| CHRIS | 0.135 | 4.52 | 4165-4800 | 780/100 sec |

Table 3: Instrumental characteristics of the LHR, CHRIS, EM27/SUN and IFS125HR of TCCON.

The methodology from Section 3.2 is applied: the state vector includes only $CO_2$ concentrations across 0–40 km layers, incorporating the SNR and spectral resolution specific to the FTS instruments (see **Table 3**). A comparison of averaging kernels (cf. El Kattar, Auriol and Herbin, 2020) with FTS instruments reveals sharper peaks and a more homogeneous vertical distribution than CHRIS, EM27/SUN and IFS125HR, suggesting higher sensitivity at higher altitudes though the a posteriori error $S_x$ is significantly reduced in the lower atmosphere. This is further supported by the error budget analysis: the a posteriori total error (solid black line) remains distinguishable from the a priori error (red line) even in the higher atmosphere as seen in the right panel of **Figure 3**. This discrepancy is due to LHR's higher spectral resolution compared to FTS instruments, ensuring continuous enhancement of our understanding along the atmospheric column.

**Table 4** shows the DOFs for $CO_2$ and the total profile error for both viewing angles using the diagonal a priori covariance matrices. FTS instruments have DOFs ranging from 2.95 at 10° to 4.23 at 80°, while for the LHR, they're 4.13 and 5.15 respectively. This means the LHR can retrieve the same number of $CO_2$ partial columns at 10° and an extra column at 80° compared to the FTS instruments. Generally, at an 80° angle, the LHR can retrieve one to two additional $CO_2$ columns in the troposphere, while the profile error remains the same at 10° and improves at 80°.

|  | DOFs | | Error | |
|---|---|---|---|---|
|  | 10° | 80° | 10° | 80° |
| LHR | 2.79 | 2.4 | 2.4% | 1.95% |
| EM27/SUN* | 2.37 | 2.68 | 1.01% | 0.97% |
| IFS125HR (TCCON)* | 3.28 | 3.53 | 0.97% | 0.95% |
| CHRIS* | 2.38 | 3.08 | 1.01% | 0.94% |

Table 4: DOFs and column errors (%) for $CO_2$, per instrument and viewing angle. *from the previous study (El Kattar, Auriol, and Herbin 2020)

# 7 Channel selection

The time required to obtain one spectrum with the LHR depends on the chosen spectral range and step which can take a long time (up to 15 mins). To optimize acquisition, we preselect the most informative spectral points, hereafter referred to as channels, prior to measurement. Each channel corresponds to an individual wavenumber bin in the radiance spectrum. This selection reduces the acquisition time and allows more spectra to be collected which can lead to better daily statistics while comparing with satellites. Furthermore, using all channels in retrieval significantly increases computational time and

systematic errors due to species correlation, complicating the evaluation of the a priori state vector $x_a$ and the error covariance matrix $S_a$. Channel selection, as described by Rodgers (2000), optimizes retrievals by identifying the subset of channels offering the most information from high-resolution infrared sounders. Cooper et al. (2006) and Kuai et al. (2010) present a depiction of this process rooted in the Shannon information content which we describe in this section.

Firstly, an "information spectrum" is constructed to assess the information content concerning the a priori state vector. The

channel with the highest information content is selected, and the a posteriori covariance matrix is updated to include its contribution. Using this updated state space, a second channel is chosen to maximize information relative to the new covariance matrix. This iterative process continues until the remaining channels' information falls below the measurement noise level. As suggested by Shannon information content and Rodgers (2000), it is beneficial to work in a basis where measurement errors and prior variances are uncorrelated, enabling comparison of measurement error with prior variability. Thus, the Jacobian

matrix $K$ (see Sect. 3.2), including the baseline, is transformed into $\widetilde{K}$ using:

$$\widetilde{K} = S_y^{-1/2} K S_a^{1/2}, \tag{14}$$

where both the a priori and measurement covariance matrices are unit matrices. Rodgers also shows that the number of singular values of $\widetilde{K}$ greater than unity determines the effective rank of the problem, representing independent measurements exceeding the noise measurement.

Let $S_i$ represent the error covariance matrix for the state space after $i$ channels have been selected. The information content of channel $j$ among the remaining unselected channels is expressed as:

$$H_j = \frac{1}{2} log_2 (1 + \widetilde{k_j^T} S_i \widetilde{k_j}), \tag{15}$$

where $\widetilde{k_j}$ is the $j$th row of $\widetilde{K}$. $H_j$ represents the information spectrum (expressed in bits), used to select the first channel. If channel $l$ is chosen, the covariance matrix is updated for the next iteration using:

$$S_{i+1}^{-1} = S_i^{-1} + \widetilde{k_l}\widetilde{k_l^T}, \tag{16}$$

Channels are selected iteratively until 90% of the total information spectrum $H$ is achieved, ensuring the measurement noise threshold is not exceeded.

| DOFs | $CO_2$ |
| --- | --- |

|  | 90% | 99% |
|---|---|---|
| Number of channels | 408 | 1919 |
| Percentage of the total number of channels | 8.24% | 38.78% |

Table 5: Number of selected channels for the DOFs of $CO_2$ and their percentage of the total channels for the LHR.

After converting $H$ to DOFs, we obtained **Figure 4**, which illustrates the evolution of the $CO_2$ total DOFs as a function of the number of selected channels for a SZA of 10°. Initially, in **Figure 4**, the DOFs show a sharp increase with the first selected channels, followed by a more gradual rise. **Table 5** presents the number of channels needed to achieve 90% and 99% of the total information. Out of the 4949 exploitable channels in the LHR, only 8.24% (408 channels) are necessary to reach 90% of the retrieved information and 38.78% are needed for 99% of the information. In other words, using selected channels corresponding to 90% of the total information content produces results comparable to using all channels, as nearly 92% of the information is redundant.

Additionally, in **Figure 5**, we present the first 100 selected channels ranked by their information content with respect to our Jacobian. The first 30 channels are shown in red, channels 31 to 60 in blue, and channels 61 to 100 in green. Notably, the information is primarily concentrated around three absorption lines in the range 6362-6365 cm$^{-1}$. Interestingly, nearly 30% of the top 100 channels lie in baseline regions with little to no $CO_2$ absorption. This suggests that, in future acquisitions, the combined range can be used to enable faster measurements while preserving a small scan step. These results emphasize the importance of identifying the best channels for $CO_2$, making the retrieval process easier and more efficient, which is one of

the benefits of using a broadband tunable laser. We consider this finding to be one of the most significant outcomes of this study.

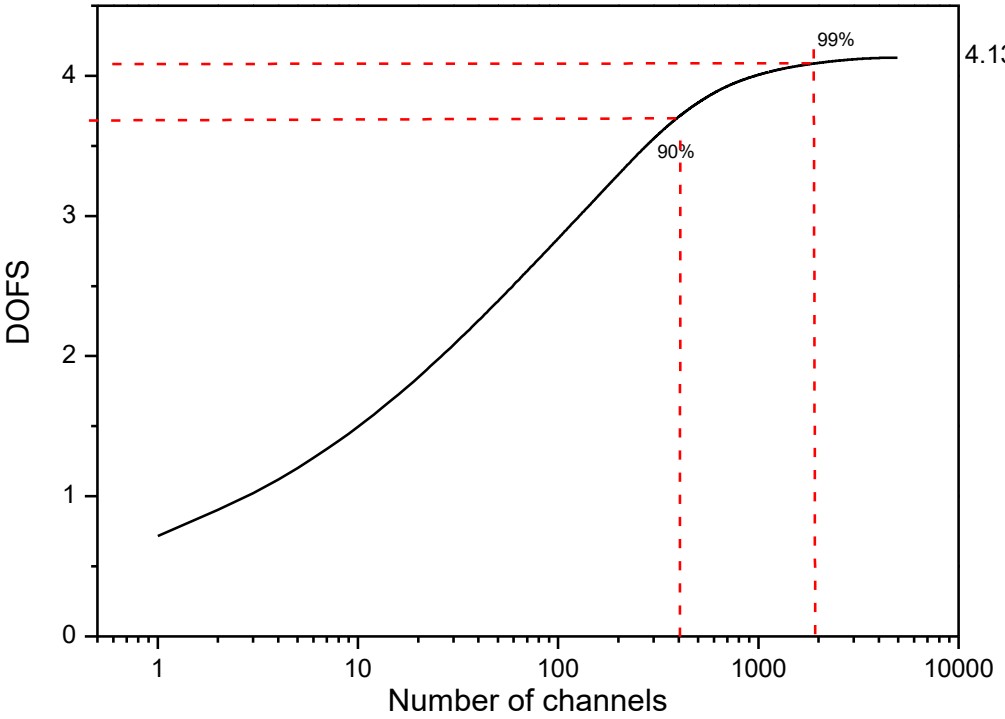

 Figure 4: Evolution of the DOFs with the number of selected channels for $CO_2$.

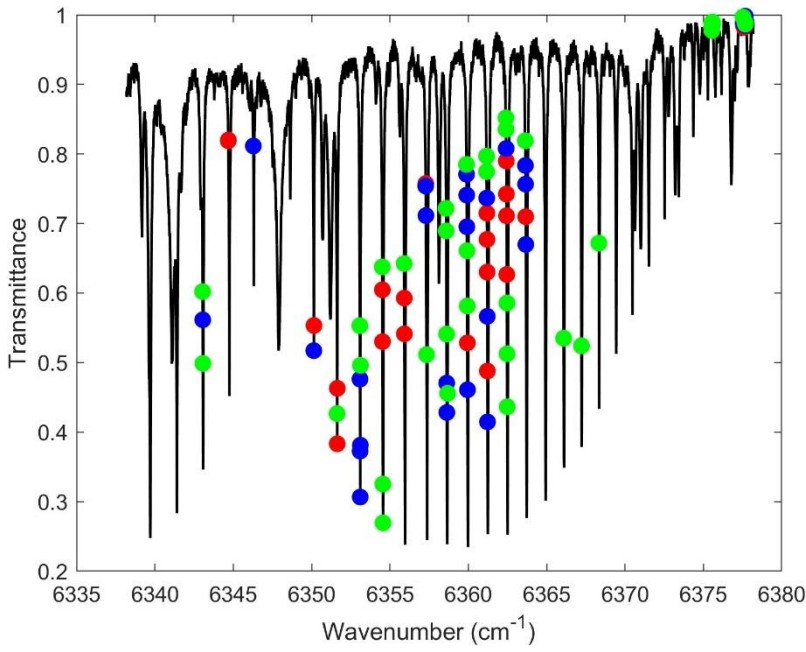

Figure 5: Micro-window selection for $CO_2$ retrieval. The first 30 channels shown in red, channels 31 to 60 in blue, and channels 61 to 100 in green. These channels are ranked based on their information content with respect to the $CO_2$ Jacobian.

## 8 Conclusions

In conclusion, this paper presents the measurement capabilities of a new near-infrared laser heterodyne radiometer, which allows the retrieval of $CO_2$ in the atmospheric column based on ground heterodyne measurement of the sunlight. This spectro-radiometer has an exceptionally high spectral resolution ($0.0047$ cm$^{-1}$) and an exploitable spectral domain ranging from 6338 to 6378 cm$^{-1}$. An extensive information content analysis is conducted to evaluate the LHR's potential for $CO_2$ retrieval, using two SZAs ($10°$ and $80°$) to quantify the impact of solar optical path on information quality. The integrated profile uncertainty

is estimated, revealing a 2.74% error at $10°$ with a diagonal a priori covariance matrix and 1.72% when using a non-diagonal covariance matrix. Furthermore, a comparison has been carried out with the referenced FTS instruments, such as TCCON's IFS125HR and COCCON's EM27/SUN, both widely utilized in satellite validation. The LHR exhibits unique advantages in retrieving gas columns with better vertical discretization. It is therefore a promising complementary instrument for local scale measurements or for satellite validation. Finally, a channel selection is implemented to eliminate redundant information and

identify an optimal spectral range to improve daily statistics.

## Author contributions

TW and AS performed the measurements; MTEK analyzed the data and wrote the manuscript draft; HH and WC validated, reviewed and edited the manuscript.

## Competing interests

The authors declare that they have no conflict of interest.

## Data availability

All data used in this study are publicly available. The ERA5 reanalysis data were obtained from the Copernicus Climate Data Store (https://cds.climate.copernicus.eu/). The AirCore data were obtained from the publicly available repository (https://data.ipsl.fr/repository/MAGIC/). Arahmis and the custom MATLAB codes used for the retrieval analysis is available 395 from the corresponding author upon reasonable request.

## Acknowledgments

The authors thank the co-financial supports from the LABEX CaPPA project (ANR-10-LABX005), the CPER ECRIN program, the European Union's Horizon 2020 Research and Innovation Program under the Marie Skłodowska-Curie grant agreement No 872081.

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
