# Peer review of "Potential CO2 measurement capabilities of a transportable Near Infrared Laser Heterodyne Radiometer (LHR)"

_EGUsphere, 2025_

## Referee Comment (RC2)

**Review of "Potential CO$_2$ measurement capabilities of a transportable Near Infrared Laser Heterodyne Radiometer (LHR) (AMT-2025-250)"**

The manuscript presents an innovative approach for measuring atmospheric CO$_2$ using a portable Laser Heterodyne Radiometer (LHR) operating in the near-infrared (NIR) region. This work is highly relevant to the scope of Atmospheric Measurement Techniques (AMT), and the manuscript provides a thorough description of the experimental setup, along with a theoretical framework for information content analysis. The authors apply this framework to quantify the Degrees of Freedom (DOF) of the LHR instrument and compare its performance against established systems such as the Total Carbon Column Observing Network (TCCON) and the COllaborative Carbon Column Observing Network (COCCON).

While the scientific content is of interest, the organization of the manuscript would benefit from improvement to enhance the logical flow and clarity for the reader. Several key concepts require more detailed explanation to ensure they are accessible to a broader audience. Below are some general comments that I believe would help strengthen the manuscript:

1. I suggest restructuring Section 3 to better reflect the logical progression of the work, which currently mixes theoretical background, instrument-specific inputs, and results into a single extended section. Reorganizing the content under a clearer functional structure:
   — **Theory → Application → Results → Comparison** —would significantly improve clarity and readability. Specifically:

   **Theory:** Sections 3.1 and 3.2 present the forward model and the framework for information content analysis. These form the core theoretical basis of the study and could be grouped together under a dedicated section on theory.

   **Application:** Section 3.3 introduces the a priori information, measurement error covariance, and uncertainties in non-retrieved parameters as they pertain to the LHR. This section represents the application of the theoretical framework to the specific case of the LHR instrument and should be distinguished from the more abstract theory above.

   **Results:** Section 3.4, which applies the framework to retrieve CO$_2$ information content and uncertainty from LHR simulations, could be promoted to its own section—e.g., "Information Content and Uncertainty Estimation for the LHR". This would clearly signal the shift to presenting results derived from the defined retrieval setup.

   **Comparison:** Section 3.5, which compares the LHR with existing systems like TCCON and COCCON, should also be elevated to a standalone section, such as "Comparison with Existing Networks", to help readers easily locate this critical performance evaluation.

   The current placement of Sections 3.3.2 and 3.3.3 under the heading "A priori information" may be misleading. While Section 3.3.1 appropriately discusses the CO$_2$ profile and its covariance matrix as part of the a priori state vector, Section 3.3.2 refers to measurement error, and Section 3.3.3 introduces parameters such as temperature, humidity, and SZA as non-retrieved. However, in many retrieval frameworks, temperature and humidity profiles are typically treated as a priori inputs. Please clarify your definition of "a priori" to avoid confusion regarding the role of these parameters in the forward model versus the retrieval.

2. In the introduction (lines 30–36), I recommend expanding the description of the EM27/SUN spectrometer to improve clarity. For example, line 31 should clearly refer to it as the Bruker

EM27/SUN, and you can also include the spectral resolution for comparison against the earlier stated IFS125HR spectral resolution.

Additionally, the statement "the drawback of being portable is that the FTS reduces the spectral resolution" is somewhat misleading. The reduced resolution is not inherently due to portability but rather results from design trade-offs in optical path difference: smaller instruments have shorter maximum optical path lengths, which limits achievable resolution.

You could also expand on the consequences of lower spectral resolution. Specifically, lower resolution can limit the ability to resolve narrow absorption lines, potentially leading to increased interference from neighboring lines, reduced retrieval precision, and sensitivity to pressure broadening effects.

Moreover, there are published studies that directly compare the performance of the IFS125HR and the EM27/SUN, such as Herkommer et al. (2024) and Mostafavi Pak et al. (2023), which show that $CO_2$ retrievals from the EM27/SUN differ by only approximately 0.1%, which is quite impressive given its lower spectral resolution. This raises an important question for the present study: does the LHR system, with its much higher spectral resolution, offer a meaningful improvement over this offset?

3. In Section 3.1, where you describe the use of PTU Vaisala radiosondes and ancillary data from the TCCON database, I suggest adding more specific information to improve transparency and reproducibility.

   For the PTU Vaisala radiosonde, please include the typical accuracy specifications for temperature, pressure, and relative humidity. These values are likely used to define the uncertainties in your forward model or retrievals later in the analysis (e.g., Section 3.3.3), so it would be helpful to establish them clearly at this stage.

   Regarding the TCCON database, it would be beneficial to specify which TCCON station the ancillary $CO_2$ and $H_2O$ data are derived from, especially considering the measurements are conducted in Dunkirk (51.035°N, 2.369°E). Are you using a nearby TCCON site (e.g., Orléans)? Additionally, please clarify what do you mean by ancillary data? Do you mean the a priori profiles?

4. In Section 3.5, you describe differences in averaging kernels between the LHR and existing FTS instruments (e.g., EM27/SUN and IFS125HR). To support this comparison more effectively, I recommend including a plot with the averaging kernels from those FTS instruments overlaid on top of the LHR kernel.

5. In Section 4, the term "channel selection" is used to describe the identification of individual absorption lines with the highest information content. However, in the TCCON and EM27/SUN communities, "channel" typically refers to detector channels (e.g., InGaAs vs. Si), rather than specific spectral lines or intervals within an absorption band. This difference in terminology may lead to confusion for readers familiar with those systems. To improve clarity, consider using more precise terms such as "line selection" or "micro-window selection", or alternatively, explicitly define your use of "channel" at the beginning of the section.

6. In the conclusion, you report a 2.74% error in total column $CO_2$ at 10° SZA. This level of uncertainty appears quite high, especially when compared to existing ground-based systems: TCCON reports an error budget of 0.16% for $XCO_2$, and the COCCON network shows an average offset of 0.1% relative to TCCON (e.g., Herkommer et al., 2024; Mostafavi Pak et al., 2023).

   Given that one of the key motivations stated in the introduction is that LHR's higher spectral resolution should improve retrieval quality, the reported uncertainty seems to contradict this expectation. It would be important to clarify how this instrument would compete with EM27/SUN in operational or satellite-validation contexts.

**Minor corrections and comments:**

- **Line 9:** Please be more specific, what type of sensitivity you are referring to. What kind of resolution is meant, spectral, temporal, vertical?

- **Line 16:** ... an extensive analysis...

- **Line 36:** "heterodyne spectro-radiometer" is not a method, maybe you mean measurement technique?

- **Line 52-71:** You seem to be switching from the present tense (Solar radiation is captured ...) to the past tense (The modulated radiation was split by ...). I recommend using the present tense throughout, since this is a description of the standard setup.

- **Line 121-122:** The variables A and $S_x$ are introduced before they are defined in equations 3 and 5. Please consider restructuring the paragraphs accordingly.

- **Line 158:** "The a priori error covariance matrix $S_a$ can be evaluated using in-situ data or climatology, but diagonal matrices are often used for space-based retrievals." The use of "but" in the sentence implies a contrast that does not really exist.

- **Line 163:** Define perr.

- **Line 198:** please be more specific what do you mean by Kernels. do you mean posterior(total), measured, etc?

- **Line 232:** The sentence "the total column uncertainty is calculated by adding up the concentration of each layer, adjusted by the dry air column (Figure 3)" is unclear and may be misleading. Summing layer concentrations gives the total column amount, but uncertainty in the total column requires proper error propagation.

- **Line 235:** the term OPD appears to be misused. If you are referring to the increased atmospheric path length at high solar zenith angles, "slant path" would be the correct terminology.

- **Line 251:** By green line and violet line, it seems like you are referring to Figure 3, please mention it.

- **Table 1:** I recommend adding a more comprehensive caption that explains what each state vector element refers to (e.g., whether $CO_2$ refers to a profile or total column scaling). Be more specific about what you mean by TCCON database.

- **Figure 2:** please clearly indicate which curve corresponds to the measured LHR spectrum and which one to the ARAHMIS simulation. In addition, could you clarify why $CH_4$ was not included in the forward model simulation shown? I would also recommend adding a residual plot (i.e., measured – modeled) below the main panel.

- **Table 2:** why is SZA = 10° used as the minimum value? At your measurement site in Dunkirk, the lowest achievable SZA is around 30° in summer. Using a range like 30°–80° would be more realistic and representative of actual observing conditions.

- **Table 3:** you present the full spectral ranges of the EM27/SUN and IFS125HR instruments. However, it would be more informative to also include the specific $CO_2$ micro-windows typically used for retrievals with these instruments. This would allow for a more direct and meaningful comparison with the spectral region covered by the LHR.

- **Figure 3:** The left panel displays numerous colored lines representing averaging kernels, but the caption and legend do not explain what these colors signify. Additionally, the right panel legend includes five color-coded components, but do they apply to the left panel? Please consider separating or clarifying the legends to avoid ambiguity. In addition, please explicitly state in the caption that the figure corresponds to a solar zenith angle (SZA) of 10°.

---

## Author Comment (AC1)

**Authors response to the reviewer**

The manuscript presents an innovative approach for measuring atmospheric $CO_2$ using a portable Laser Heterodyne Radiometer (LHR) operating in the near-infrared (NIR) region. This work is highly relevant to the scope of Atmospheric Measurement Techniques (AMT), and the manuscript provides a thorough description of the experimental setup, along with a theoretical framework for information content analysis. The authors apply this framework to quantify the Degrees of Freedom (DOF) of the LHR instrument and compare its performance against established systems such as the Total Carbon Column Observing Network (TCCON) and the Collaborative Carbon Column Observing Network (COCCON).

While the scientific content is of interest, the organization of the manuscript would benefit from improvement to enhance the logical flow and clarity for the reader. Several key concepts require more detailed explanation to ensure they are accessible to a broader audience. Below are some general comments that I believe would help strengthen the manuscript:

1. I suggest restructuring Section 3 to better reflect the logical progression of the work, which currently mixes theoretical background, instrument-specific inputs, and results into a single extended section. Reorganizing the content under a clearer functional structure: — **Theory → Application → Results → Comparison** —would significantly improve clarity and readability. Specifically:

   **Theory:** Sections 3.1 and 3.2 present the forward model and the framework for information content analysis. These form the core theoretical basis of the study and could be grouped together under a dedicated section on theory.

   **Application:** Section 3.3 introduces the a priori information, measurement error covariance, and uncertainties in non-retrieved parameters as they pertain to the LHR. This section represents the application of the theoretical framework to the specific case of the LHR instrument and should be distinguished from the more abstract theory above.

   **Results:** Section 3.4, which applies the framework to retrieve $CO_2$ information content and uncertainty from LHR simulations, could be promoted to its own section—e.g., "Information Content and Uncertainty Estimation for the LHR". This would clearly signal the shift to presenting results derived from the defined retrieval setup.

   **Comparison:** Section 3.5, which compares the LHR with existing systems like TCCON and COCCON, should also be elevated to a standalone section, such as "Comparison with Existing Networks", to help readers easily locate this critical performance evaluation.

   The current placement of Sections 3.3.2 and 3.3.3 under the heading "A priori information" may be misleading. While Section 3.3.1 appropriately discusses the $CO_2$ profile and its covariance matrix as part of the a priori state vector, Section 3.3.2 refers to measurement error, and Section 3.3.3 introduces parameters such as temperature, humidity, and SZA as non-retrieved. However, in many retrieval frameworks, temperature and humidity profiles are typically treated as a priori inputs. Please clarify your definition of "a priori" to avoid confusion regarding the role of these parameters in the forward model versus the retrieval.

2.  In the introduction (lines 30–36), I recommend expanding the description of the EM27/SUN spectrometer to improve clarity. For example, line 31 should clearly refer to it as the Bruker EM27/SUN, and you can also include the spectral resolution for comparison against the earlier stated IFS125HR spectral resolution.

    Additionally, the statement "the drawback of being portable is that the FTS reduces the spectral resolution" is somewhat misleading. The reduced resolution is not inherently due to portability but rather results from design trade-offs in optical path difference: smaller instruments have shorter maximum optical path lengths, which limits achievable resolution.

    You could also expand on the consequences of lower spectral resolution. Specifically, lower resolution can limit the ability to resolve narrow absorption lines, potentially leading to increased interference from neighboring lines, reduced retrieval precision, and sensitivity to pressure broadening effects.

    Moreover, there are published studies that directly compare the performance of the IFS125HR and the EM27/SUN, such as Herkommer et al. (2024) and Mostafavi Pak et al. (2023), which show that $CO_2$ retrievals from the EM27/SUN differ by only approximately 0.1%, which is quite impressive given its lower spectral resolution. This raises an important question for the present study: does the LHR system, with its much higher spectral resolution, offer a meaningful improvement over this offset?

3.  In Section 3.1, where you describe the use of PTU Vaisala radiosondes and ancillary data from the TCCON database, I suggest adding more specific information to improve transparency and reproducibility.

    For the PTU Vaisala radiosonde, please include the typical accuracy specifications for temperature, pressure, and relative humidity. These values are likely used to define the uncertainties in your forward model or retrievals later in the analysis (e.g., Section 3.3.3), so it would be helpful to establish them clearly at this stage.

    Regarding the TCCON database, it would be beneficial to specify which TCCON station the ancillary $CO_2$ and $H_2O$ data are derived from, especially considering the measurements are conducted in Dunkirk (51.035°N, 2.369°E). Are you using a nearby TCCON site (e.g.,Orléans)? Additionally, please clarify what do you mean by ancillary data? Do you mean the a priori profiles?

4.  In Section 3.5, you describe differences in averaging kernels between the LHR and existing FTS instruments (e.g., EM27/SUN and IFS125HR). To support this comparison more effectively, I recommend including a plot with the averaging kernels from those FTS instruments overlaid on top of the LHR kernel.

5.  In Section 4, the term "channel selection" is used to describe the identification of individual absorption lines with the highest information content. However, in the TCCON and EM27/SUN communities, "channel" typically refers to detector channels (e.g., InGaAs vs. Si), rather than specific spectral lines or intervals within an absorption band. This difference in terminology may lead to confusion for readers familiar with those systems. To improve clarity, consider using more precise terms such as "line

selection" or "micro-window selection", or alternatively, explicitly define your use of "channel" at the beginning of the section.

6. In the conclusion, you report a 2.74% error in total column $CO_2$ at 10° SZA. This level of uncertainty appears quite high, especially when compared to existing ground-based systems:
   TCCON reports an error budget of 0.16% for $XCO_2$, and the COCCON network shows an average offset of 0.1% relative to TCCON (e.g., Herkommer et al., 2024; Mostafavi Pak et al., 2023).
   Given that one of the key motivations stated in the introduction is that LHR's higher spectral resolution should improve retrieval quality, the reported uncertainty seems to contradict this expectation. It would be important to clarify how this instrument would compete with EM27/SUN in operational or satellite-validation contexts.

**Response to Reviewer Comments**

We thank the reviewer for their thorough and insightful review. We appreciate the recognition of the novelty and relevance of our approach, as well as the constructive suggestions to improve the manuscript. Below, we respond point-by-point to the comments and outline the corresponding revisions made.

1. Reorganization of Section 3

We have reorganized Section 3, as the reviewer asked, as follows:
- Section 3: Theoretical Framework, now includes the forward model and information content analysis (3.1 and 3.2).
- Section 4: Application to the LHR Instrument, now includes the specifics of the a priori state, measurement errors, and non-retrieved parameter treatment (revised from 3.3).
- Section 5: Results → Information Content and Uncertainty, contains the analysis based on LHR simulations (previously 3.4).
- Section 6: Comparison with Existing Networks, presents the comparison with TCCON and COCCON systems (previously 3.5).

We have also clarified the use of the term "a priori" in Section 4.1. In our revised manuscript, we now define this term more precisely to include parameters such as temperature and humidity profiles that are not retrieved but are incorporated as input into the forward model with associated uncertainties. These inputs contribute to the total error budget and are treated using an ensemble of perturbations, as clarified in Section 4.3.

2. Clarification of EM27/SUN description and spectral resolution

We appreciate the reviewer's suggestions regarding the description of the EM27/SUN spectrometer:

- In the Introduction (lines 30–36), we now explicitly refer to the Bruker EM27/SUN, and include its nominal spectral resolution of $0.5$ cm$^{-1}$, in contrast to the IFS125HR's $0.02$ cm$^{-1}$.
- We have revised the sentence about portability and spectral resolution to clarify that reduced resolution arises from design trade-offs in optical path length due to compactness, not portability per se.
- We now cite Herkommer et al. (2024) and Mostafavi Pak et al. (2023) to highlight that the EM27/SUN still performs remarkably well in $CO_2$ retrievals. Please refer to answer 6 to reflect on whether the increased resolution of LHR leads to meaningful improvements in retrieval accuracy.

3. Radiosonde accuracy and ancillary data clarification

We have expanded the description in Section 3.1 (now Section 3) as follows:

- For the PTU Vaisala radiosonde (PTU300), we now provide typical manufacturer-specified uncertainties: ±0.2°C (temperature), ±0.3 hPa (pressure), and ±1% RH. These values are referenced and used to estimate perturbations in temperature and humidity profiles for the uncertainty analysis in Section 4.3.
- We clarify that ancillary data refers to a priori profiles of $CO_2$ and $H_2O$ used to construct the state vector and prior covariance matrix. In our case, these are derived respectively from the AirCore launches from the MAGIC campaigns and the Orléans TCCON station, which is the closest operational site to Dunkirk from 2016 to 2023.

4. Averaging Kernel comparison plot

We agree that a direct visual comparison would enhance the interpretation of our results. However, overlaying the averaging kernels significantly reduces the clarity of the figure, as more than 160 lines become indistinguishable. Therefore, we refer the reader to our previous study for a detailed comparison of these averaging kernels.

5. Terminology clarification on "Channel Selection"

To avoid confusion with terminology used in the TCCON and EM27/SUN communities, we have now explicitly defined the term "channel" at the beginning of Section 7 (previously Section 4). In this study, "channel" refers to an individual spectral point (i.e., a specific wavenumber bin) in the measured radiance spectrum. We have also updated the caption of Figure 5 to reflect this definition and added the term "micro-window selection" where appropriate to clarify that this selection is based on information content per spectral point.

6. Reported $XCO_2$ uncertainty and comparability to TCCON/COCCON

We fully agree that the current level of uncertainty appears high compared to the operational performance of mature networks such as TCCON and COCCON. However, we would like to clarify that the reported 2.74% corresponds to the vertically integrated profile retrieval uncertainty, not to a total column $XCO_2$ uncertainty derived from a ratio of $CO_2$ and $O_2$ columns as in TCCON/COCCON. Since our current setup does not yet include an $O_2$ channel (due to the lack of a suitable laser source in the 1.26 μm region), a true $XCO_2$ product cannot yet be derived. For this reason, and to avoid confusion, we have renamed the reported quantity "integrated profile uncertainty" in the revised manuscript.

We agree that the high spectral resolution of the LHR holds great potential to reduce smoothing errors and improve retrieval quality. A full profile retrieval for $CO_2$ is currently under development and will be presented in a future study. We expect that this, combined with the future addition of an $O_2$ channel, will enable a direct and fair comparison with TCCON/COCCON $XCO_2$ error budgets, including potential advantages in vertical sensitivity.

In this study, we focus on the initial demonstration of information content and error propagation for a profile retrieval from a compact LHR instrument, while acknowledging that further development is needed before it can match or surpass operational standards for satellite validation.

**Minor corrections and comments:**

**• Line 9: Please be more specific, what type of sensitivity you are referring to. What kind of resolution is meant, spectral, temporal, vertical?**

**Response:** We have revised this sentence to clarify that we are referring specifically to *vertical sensitivity* enabled by the high *spectral resolution* of the LHR instrument.

**• Line 16: ... an extensive analysis...**

**Response:** Corrected as suggested.

**• Line 36: "heterodyne spectro-radiometer" is not a method, maybe you mean measurement technique?**

**Response:** We agree and have changed "heterodyne spectro-radiometer" from being described as a method to "measurement technique" for accuracy.

**• Line 52-71: You seem to be switching from the present tense (Solar radiation is captured ...) to the past tense (The modulated radiation was split by ...). I recommend using the present tense throughout, since this is a description of the standard setup.**

**Response:** We have revised this section to consistently use the present tense.

• **Line 121-122: The variables A and Sx are introduced before they are defined in equations 3 and 5. Please consider restructuring the paragraphs accordingly.**

**Response:** We have restructured the text to ensure that variables $A$ (averaging kernel matrix) and $S_x$ (posterior error covariance matrix) are first introduced conceptually before being formally defined in Equations 3 and 5, respectively.

• **Line 158: "The a priori error covariance matrix Sa can be evaluated using in-situ data or climatology, but diagonal matrices are often used for space-based retrievals." The use of "but" in the sentence implies a contrast that does not really exist.**

**Response:** The sentence has been revised especially since we add a part where we use an a priori covariance matrix.

• **Line 163: Define perr.**

**Response:** The variable $p_{err}$ is now defined.

• **Line 198: please be more specific what do you mean by Kernels. do you mean posterior(total), measured, etc?**

**Response:** We now clarify that "Kernels" refers specifically to *averaging kernels* associated with the posterior solution, calculated via Equation 4. This clarification has been added in the revised text.

• **Line 232: The sentence "the total column uncertainty is calculated by adding up the concentration of each layer, adjusted by the dry air column (Figure 3)" is unclear and may be misleading. Summing layer concentrations gives the total column amount, but uncertainty in the total column requires proper error propagation.**

**Response:** The sentence has been revised (see answer 6).

• **Line 235: the term OPD appears to be misused. If you are referring to the increased atmospheric path length at high solar zenith angles, "slant path" would be the correct terminology.**

**Response:** "OPD" was misused here. We have replaced "OPD" with atmospheric path length at higher SZA.

• **Line 251: By green line and violet line, it seems like you are referring to Figure 3, please mention it.**

**Response:** We have revised the sentence to explicitly reference Figure 3.

**• Table 1: I recommend adding a more comprehensive caption that explains what each state vector element refers to (e.g., whether $CO_2$ refers to a profile or total column scaling). Be more specific about what you mean by TCCON database.**

**Response:** We have expanded the caption to clarify:
- Whether each state vector element refers to a profile or scaling factor (e.g., $CO_2$ is a profile, SZA is a scalar).
- That "TCCON database" refers to publicly available Level 2 products from the Orléans station, which were used to define a priori $CO_2$ and $H_2O$ profiles.

**• Figure 2: please clearly indicate which curve corresponds to the measured LHR spectrum and which one to the ARAHMIS simulation. In addition, could you clarify why $CH_4$ was not included in the forward model simulation shown? I would also recommend adding a residual plot (i.e., measured – modeled) below the main panel.**

**Response:**
- We now label which curve corresponds to the measured LHR spectrum and which corresponds to the ARAHMIS simulation.
- $CH_4$ was excluded from the forward model in this case for clarity and because its absorption lines do not overlap with the selected $CO_2$ micro-window.
- As suggested, a residual plot (observed – calculated) has been added below the main panel to help visualize the fit quality.

**• Table 2: why is SZA = 10° used as the minimum value? At your measurement site in Dunkirk, the lowest achievable SZA is around 30° in summer. Using a range like 30°–80° would be more realistic and representative of actual observing conditions.**

**Response:** We agree that a 10° solar zenith angle is rather unrealistic for high-quality direct sun observations. In our study, the 10° (and subsequently 80°) cases are used primarily as theoretical scenario to demonstrate the two extremes of the instrument's operating range, rather than to represent typical observation conditions. Our aim was to explore the range of sensitivity under idealized geometries and to facilitate comparison with previous work.

**• Table 3: you present the full spectral ranges of the EM27/SUN and IFS125HR instruments. However, it would be more informative to also include the specific $CO_2$ micro-windows typically used for retrievals with these instruments. This would allow for a more direct and meaningful comparison with the spectral region covered by the LHR.**

**Response:** We have now added a new row to the table listing the typical $CO_2$ micro-windows used in retrievals for the EM27/SUN (6173 to 6390 $cm^{-1}$) and the IFS125HR (6300 $cm^{-1}$ band). This allows for a clearer comparison with the spectral region targeted by the LHR.

**• Figure 3: The left panel displays numerous colored lines representing averaging kernels, but the caption and legend do not explain what these colors signify. Additionally, the right**

**panel legend includes five color-coded components, but do they apply to the left panel? Please consider separating or clarifying the legends to avoid ambiguity. In addition, please explicitly state in the caption that the figure corresponds to a solar zenith angle (SZA) of 10°.**

**Response:**

- We revised the legend and caption to clearly explain the color coding of the left panel, which shows the averaging kernels for each retrieval level.
- We clarified that the right panel's legend applies only to the uncertainty decomposition.
- The caption now explicitly states that the figure corresponds to a solar zenith angle (SZA) of 10°.

---

## Author Comment (AC2)

**Authors response to the reviewer**

We would like to thank the reviewer for their careful reading and thoughtful evaluation of our manuscript. We are encouraged by the recognition of the significance of our work, especially the potential of using a widely tunable diode laser in laser heterodyne radiometry (LHR) for atmospheric remote sensing. We also appreciate the positive comments regarding the information content analysis, and channel selection strategy. In response to the reviewer's suggestions, we have revised the manuscript to improve clarity, particularly in terms of the retrieval setup, the definition of state and measurement vectors and the role of background channels in the information content analysis. We have also expanded acronym definitions and addressed the technical questions and semantic points raised in both the general and line-by-line comments. Our aim throughout the revision has been to improve the accessibility of the manuscript for both atmospheric remote sensing specialists and those coming from a laboratory spectroscopy background.

**Reviewer Comment: Many acronyms are not explained (TCCON, COCCON, MAGIC, FORUM). With some this might be fine (i.e., citation to the network main paper), but with others not.**

**Response:** We have revised the manuscript to ensure that all acronyms are defined at first mention. We also provide, where relevant, references to foundational publications describing these networks/missions.

**Reviewer Comment: You write a lot about your Model and Retrieval, but in the end it is a bit vague to me what exactly you use in your state vector (for the gases: mixing ratios, concentrations, column densities) and in your measurement vector (radiances, transmittance, ...) - I would consider this the most relevant information on a higher level of how your retrieval is designed.**

**Response:**

To address this point, we have added a paragraph early in the retrieval section that clearly defines:

- The state vector, which includes the vertical profile of $CO_2$ volume mixing ratios (VMR) on a fixed grid. Depending on the scenario, we may also retrieve scaling factors for temperature profiles, and in some cases, interfering species such as $H_2O$.

- The measurement vector consists of calibrated radiance spectra derived from observations. These are calculated by multiplying the solar spectrum measured during atmospheric absorption with the SOLAR-ISS spectrum (see response to comment on Line 94).

These clarifications are now explicitly included in Section 3.2 of the manuscript for better illustration of the retrieval components.

**Reviewer Comment:**

**Regarding your results of the information content analysis of the spectrum (Figure 4 and 5), I am not completely convinced, since I miss a few points in the discussion:**

1. **As I understand it, you do all of this analysis in some type of absorbance space - but to get there from measured radiances, a "background channel" is definitely needed - which I do not see represented in your results.**

2. **You are only considering the $CO_2$ information at the moment, but the large advantage of using a widely tunable laser is in my opinion that you can measure full rot-vib bands and get constraints on the temperature - which is degenerated with the gas amount for a single line or a few close ones. Could you see any improvements here?**

3. **Are you proposing to simply limit the used channels in a retrieval or also to limit the measured spectral bandwidth?**

**Response:**

1.      If the "background channel" here refers to a reference radiance spectrum without $CO_2$ absorption (a clean solar background spectrum), unfortunately, such a measurement is not possible in our case because atmospheric absorption is always present along the sunlight's path. Instead, we adopted a commonly used method in solar absorption spectroscopy: we applied a baseline fitting procedure over a broad spectral window to approximate the background continuum. At the same time, we corrected for variations in sunlight and local oscillator laser intensity. This gives us a transmittance spectrum without needing a separate background measurement. Also, while a lab-based LHR system can measure its own heterodyne background, this isn't a valid replacement for the solar background.

2.      We agree that one of the major advantages of using a widely tunable diode laser is its ability to span entire rotational-vibrational bands, providing sensitivity to temperature through the shape and relative intensities of spectral lines. However, the primary aim of this paper is to introduce a new, time-efficient LHR system that can be deployed in field campaigns and achieves accurate retrievals comparable to those from FTIR systems. Unlike FTIR, the larger the spectral range we cover, the longer the acquisition time. Therefore, it's important to find a good balance between spectral coverage and acquisition duration. The integration time should not exceed 5 minutes; otherwise, the air column may become too mixed and the optical path length may vary.

3.      Our intention is to identify and prioritize informative spectral channels within the measured range to improve retrieval stability and reduce computational cost. Depending on the scenario, and specifically in campaign measurements, we propose to limit the measurement bandwidth, especially given the high integration time.

**Line 9f: "I find the first sentence a bit vague and not adding anything of value – the same could be said about many methods."**

**Response:** We have revised the opening sentence to make it more specific to our study and its context within LHR-based remote sensing. The new phrasing emphasizes the unique combination of broadband tunability and heterodyne resolution enabled by our setup, rather than making a general statement about LHR.

**Line 10f: "Semantically wrong in my opinion. Heterodyne detection is a method, not an instrument and thus can not be 'transportable' or similar."**

**Response:** We have revised the phrasing accordingly. The text now refers to the instrumental setup employing heterodyne detection as being portable, rather than attributing this characteristic to the detection method itself.

**Line 23: "I would say the 'radiative impacts on the atmosphere' of higher GHG concentrations is well sorted since a few decades and 'studying the effects of climate change' with an instrument as presented sounds to me at best like the analysis of changes in biological sources and sinks of GHGs."**

**Response:** We have rephrased this sentence to clarify that the system is intended to contribute to monitoring and understanding the spatiotemporal variations in greenhouse gas concentrations, which in turn supports the study of emission sources, sinks, and atmospheric transport.

**Line 24ff: "This is quite a generic reference with an even broader reference. If you cite the latest IPCC report for something like this, I would ask you to be more precise where in the thousands of pages you get that from."**

**Response:** We have now replaced the generic IPCC citation with a more targeted reference and section number from the most relevant IPCC report chapter (e.g., AR6 WG1 Ch. 1 & Ch. 10), which specifically discusses the role of trace gas measurements in climate monitoring.

**Line 31: "I wouldn't call the logistical requirements of COCCON 'low'. While the EM27/SUN is portable (unlike the instruments of the TCCON network), the logistics behind operating a network of them, ensuring the comparability of the instruments, etc. is quite substantial and an achievement."**

**Response:** Thank you for pointing this out. We have revised the description to acknowledge that while the instruments themselves are portable, the operation and maintenance of an extensive instrument network, like COCCON, is indeed non-trivial and requires significant coordination and effort. We now frame this differently rather than implying minimal effort.

**Line 35: "If you argue via cost effectiveness, could you give a rough number/price?"**

**Response:** We now specify that the cost of our prototype LHR system is approximately 20% of that of a typical EM27/SUN spectrometer. This estimate reflects our current setup and highlights the potential cost advantage of heterodyne detection, although we note that the final price is highly dependent on the choice of laser source and detector.

**Line 35 (continued): "What sensitivity limits are you talking about?"**

**Response:** The sensitivity limits referred to here, is the vertical sensitivity.

**Line 55: "Since lock-in amplifiers are not necessarily known to everybody in the target group of this journal, I would ask you to add half a sentence explaining why you modulate the light source with a chopper."**

**Response:** We updated the text to briefly explain the rationale: … A mechanical chopper that modulates light to enable phase-sensitive detection by the lock-in amplifier, isolating the heterodyne signal from low-frequency noise. This modulated sunlight…

**Line 68: "Can you also give details on the product concerning the 'square law detector'? I think this could help avoid misunderstandings, since many readers will think 'photodiode' when reading this in the context of this journal and the topic."**

**Response:** Indeed, this is not a photodiode, but a Schottky diode that's is used to extract the absorption signature, which is the envelope of the amplitude of the radio frequency signal. The output of such a detector is proportional to the square of the amplitude of the input beat signal.

**Line 77: "I think this equation is not clear at all, even after reading the cited reference. At a minimum, the chosen definition for SNR should be reiterated and the relevant assumptions stated (relative strengths of signals, shot noise limits, etc.)."**

**Response:** In this response, we provide a more detailed explanation of the SNR definition as reported in the cited reference, and we outline the main noise sources in laser heterodyne systems.

While we agree that a more complete treatment would improve clarity, a full discussion of the noise model goes beyond the scope of the present manuscript and will be addressed in a forthcoming technical paper. For this reason, we have opted not to include these details in the current version but now briefly mention the primary noise contributions for transparency.

Sunlight signal:

$$E_S(t) = A_S \cos(\omega_s t + \varphi_s) \qquad\qquad (1)$$

Local oscillator laser:

$$E_{LO}(t) = A_{LO} \cos(\omega_{LO} t + \varphi_{LO}) \qquad\qquad (2)$$

Total field intensity on the photodetector:

$$P = (E_S(t) + E_{LO}(t))^2 = (A_S \cos(\omega_s t + \varphi_s) + A_{LO} \cos(\omega_{LO} t + \varphi_{LO}))^2 = \frac{A_S{}^2 + A_{LO}{}^2}{2} +$$

$$\frac{A_S{}^2}{2} \cos(2\omega_s t + 2\varphi_s) + \frac{A_{LO}{}^2}{2} \cos(2\omega_{LO} t + 2\varphi_{LO}) + A_S A_{LO} \{ \cos[(\omega_s + \omega_{LO})t + \varphi_s + \varphi_{LO}] +$$

$$\cos[(\omega_s - \omega_{LO})t + \varphi_s - \varphi_{LO}] \} \qquad\qquad (3)$$

Filtering out high-frequency and DC components, the intermediate frequency (IF) signal is:

$$P_{IF} = A_S A_{LO} \cos[(\omega_s - \omega_{LO})t + \varphi_s - \varphi_{LO}] = 2\sqrt{P_S \times P_{LO}} \cos[(\omega_s - \omega_{LO})t + \varphi_s - \varphi_{LO}] \quad (4)$$

The photocurrent after photodetector is:

$$i_{IF} = 2 \frac{\eta e}{h\nu} \sqrt{P_S \times P_{LO}} \cos[(\omega_s - \omega_{LO})t + \varphi_s - \varphi_{LO}] \qquad\qquad (5)$$

The photocurrent amplitude is:

$$i_{IF} = 2 \frac{\eta e}{h\nu} \sqrt{P_S \times P_{LO}} \qquad\qquad (6)$$

**Main Noise Sources in Laser Heterodyne Radiometer**

1. Johnson noise:

    a) The thermal noise of the photodetector:

    $$< i_j^2 > = \frac{4kT_m \Delta f}{R_m} \qquad\qquad (7)$$

    b) The thermal noise of the amplifier:

    $$< i_A^2 > = \frac{4kT_A \Delta f}{R_A} \qquad\qquad (8)$$

2. Coherently detected thermal noise:

$$< i_{CDT}^2 >= \frac{4\eta^2 e^2}{hv} \delta_s P_{LO} \Delta f \qquad (9)$$

$$\delta_s = \left[ \exp\left(\frac{hv}{kT_s}\right) - 1 \right]^{-1} \qquad (10)$$

3. Laser-induced noise

   a) The relative intensity noise:

$$< i_{RIN}^2 >= R_{IN} \Delta f \left(\frac{\eta e}{hv} P_{LO}\right)^2 \qquad (11)$$

   b) The shot noise:

$$< i_s^2 >= 2e i_{DC} \Delta f = 2e \frac{\eta e}{hv} P_{LO} \Delta f \qquad (12)$$

Signal-to-noise Ratio

Based on the above analysis, the signal-to-noise ratio (SNR) of the LHR system can be expressed as:

$$SNR = \frac{}{++++} \qquad (13)$$

The LHR system was designed to operate in the shot noise–limited regime by optimizing the local oscillator laser power, such that the total system noise is dominated by LO-induced shot noise.

$$< i_s^2 > >> < i_j^2 > \; ; < i_s^2 > >> < i_A^2 > \qquad (14)$$

Therefore, the thermal noise can be ignored. Meanwhile, a balanced detector in LHR was used for heterodyne signal detection, eliminating the relative intensity noise of the local oscillator laser. Thus, the relative intensity noise is also ignored.

Therefore,

$$SNR = \frac{}{+} \qquad (15)$$

By substituting formulas (6), (9) and (12) into formula (15), SNR can be expressed as

$$SNR = \frac{1}{\Delta f} \frac{\eta P_S}{hv(1+2\eta\delta_s)} \qquad (16)$$

The signal light power received by the heterodyne system can be expressed as

$$P_s = 2T_0 hv \Delta f \delta_s \qquad (17)$$

Hence, the SNR can be expressed as

$$SNR = \frac{2T_0\eta}{2\eta + \exp\left(\frac{hv}{kT_S}\right) - 1} \quad\quad (18)$$

The final SNR at the output of the RF filter with the bandwidth $\Delta f$ and an integration time $\tau$ is given by:

$$SNR = \frac{2\eta T_0\sqrt{\Delta f\tau}}{2\eta + \exp\left(\frac{hv}{kT_S}\right) - 1} \quad\quad (19)$$

**Line 82: "You state the theoretical SNR and then introduce your measurements, but I do not find how well the actual measured SNR compares to the theoretical one."**

**Response:** The actual measured SNR is approximately 200, based on a single scan, in contrast to the FTIR measurements where multiple scans are averaged. The reduced SNR can be attributed to several factors, primarily the absence of spectral averaging. Additional contributors include suboptimal detector performance such as lower-than-expected quantum efficiency, elevated dark current, and electronic noise sources including amplifier and digitizer interference. We have now added a quantitative comparison between the theoretical and measured SNR values in the main text, along with a brief discussion to interpret the observed discrepancy. While current measurements yield a lower SNR (~200), an SNR of 700 is achievable through additional scan averaging or improved detector performance. We therefore use SNR = 700 to assess the theoretical information content under optimal conditions, which will be targeted in future measurement campaigns.

**Line 84: "'An information content study'?"**

**Response:** We have corrected this phrase.

**Line 90: "Why a Gaussian line profile? This would be rather unusual (and wrong). You should at least use Voigt, but even this is not necessarily up to current standards."**

**Response:** This is an important point. We agree that the original phrasing was misleading. In our retrievals, the absorption lines are modeled using Voigt profiles, consistent with standard spectroscopic practice. The use of a Gaussian line shape refers specifically to the Instrument Line Shape (ILS), which is convolved with the Voigt-profiled spectrum. This choice is based on the characteristics of the LHR instrument, whose optical and detection system yields a response that

is better approximated by a Gaussian. We have revised the text to clearly distinguish between the line profile and the ILS to avoid further confusion.

**Line 94: "What is the LATMOS function?"**

**Response:** We have now defined the "LATMOS function" explicitly in the text. It refers to a custom radiative transfer routine developed at LATMOS, which we used for the forward simulation of absorption spectra. This routine relies on the SOLAR-ISS spectrum, a high-resolution solar reference spectrum constructed by combining existing solar spectra with SOLAR/SOLSPEC measurements using known slit functions. SOLAR-ISS provides an accurate representation of the solar irradiance during the 2008 solar minimum, especially in the ultraviolet, visible, and infrared regions.

**Line 103f: "To make any statement about the consistency a plot of the residuals between measurement and forward model is required."**

**Figure 2: "Please be more clear in the layout and caption of the figure what is measured data and what is simulation."**

**Response:** We added a new panel for illustrating the residuals between measured and simulated spectra. The figure caption and layout have been revised to distinctly label measured data and simulated spectra as well as a clear legend for better readability.

**Line 144: "I 'I' a unity matrix?"**

**Response:** Yes, here 'I' denotes the identity matrix. We have added this information to the text.

**Line 146f: "You say that 'Sm is calculated [...]' but then proceed to give an equation for Smeas."**

**Response:** Indeed, we have corrected the notation to consistently use $S_{meas}$ in the text.

**Line 161: "I don't get the division by 100 in the equation. Looks to me like a conversion from percent to a straight number, but perror is sometimes given as absolute value including units (i.e. for the temperature) in your table."**

**Response:** In our implementation (based on ARAHMIS), the uncertainties are initially expressed in relative terms (i.e., as percentages). However, since the subsequent calculations are performed using absolute values, a conversion from percent to fractional form (i.e., division by 100) is necessary.

**Line 190: "To my understanding 10° measurements are rather unrealistic for high quality direct sun observations."**

**Response:** We agree that a 10° solar zenith angle is rather unrealistic for high-quality direct sun observations. In our study, the 10° (and subsequently 80°) cases are used primarily as theoretical scenario to demonstrate the two extremes of the instrument's operating range, rather than to represent typical observation conditions. Our aim was to explore the range of sensitivity under idealized geometries and to facilitate comparison with previous work (Kattar et al., 2020).

**Line 200f: "In a sense, this is obvious - higher up, lower pressure, less molecules in a fixed height layer. This is one of the reasons why the atmospheric retrieval codes I am familiar with use equidistant levels in pressure, which result in (roughly) equal amounts of molecules per layer. So here, it is unclear to me, how much if not all of the described effect is due to the lower number of molecules."**

**Response:** We agree that reduced pressure and molecular density at higher altitudes are major contributors to the reduced sensitivity of LHR measurements. However, this factor alone does not fully explain the effect.

At high altitudes, absorption lines become narrower due to reduced pressure broadening. These narrow lines are more difficult for instruments with finite spectral resolution to resolve, which diminishes the strength of the observed signal even when the total number of molecules is accounted for. In addition, the line strength itself can decrease due to altitude-dependent temperature effects.

Instrumental limitations such as spectral resolution, signal-to-noise ratio, and observational geometry (e.g., reduced path length through the absorbing layer) further compound the reduction in sensitivity. This behavior is consistent with satellite observations as well: for instance, in limb-viewing geometries, satellites traverse long atmospheric paths but still exhibit low sensitivity at high altitudes for similar reasons, not just low density, but also narrow line widths and weaker absorption features.

To further clarify this point, and following the editor's suggestion, we computed an information content analysis using a full covariance matrix. The results show that even in high-altitude regions with low molecular abundance, the averaging kernels can be comparable to those at lower altitudes. This suggests that reduced sensitivity cannot be attributed to molecule number alone, but results from a combination of spectral, instrumental, and geometric factors.

**Line 208: "Maybe change the order of the tables? Table 4 is needed before Table 2."**

**Response:** We added the DOFs values mentioned in Table 4 to Table 2 to improve the logical flow in the text, as Table 4 is needed in a later section.

**Figure 3: "Please add a better explanation (maybe linked to your formalism for A) what the different lines are."**

**Response:** The explanation of the different lines of A are added both in the main text and in the caption of Figure 3.

**Line 235: "Optical path difference? You are talking about a longer path in the atmosphere I assume?"**

**Response:** Yes, the term "optical path difference" refers to a longer path through the atmosphere. This expression has been clarified to avoid confusion with the instrument's optical path difference.

**Table 3: "What definition of SNR is utilized here? Is it comparable between the different measurements/works? Also: What CHRIS is remains unclear to somebody not familiar with the corresponding paper."**

**Response:** We have added the integration times alongside the SNR value in Table 3. We confirm that, with this information, the SNR is comparable to the other measurements. Additionally, the acronym CHRIS is now defined in the table caption, with a reference to the corresponding publication for readers who may be unfamiliar with it.

**Figure 5: "Axis ticks very hard to read."**

**Response:** The figure has been reformatted to improve the readability of the axis ticks. As recommended by the editor, it now includes the baseline in the Jacobian, and the spectrum is shown instead of the Jacobian for better visualization.

---

## Author Comment (AC3)

**Authors response to the editor**

We thank the editor for his detailed and constructive review of our manuscript. We appreciate your time and the insightful feedback that will help significantly improve the quality and impact of our work. Below, we provide our point-by-point responses and outline how we have revised the manuscript accordingly.

**Comments:**

**Section 3.1, line 90: I can hardly believe a Gaussian line shape is assumed for modelling the absorption lines, I guess you refer to the Voigt profile?**

**Section 3.1, line 90:** You are correct, and as the reviewer also pointed out, the Voigt profile is used to model our spectral lines, while the Instrument Line Shape (ILS) is Gaussian. The simulated spectrum is therefore the result of a convolution between the Voigt profile and this Gaussian ILS.

**Section 3.1, line 102: please rephrase: " …. while TCCON a-priori information is used for CO2 and H2O atmospheric profiles."**

**Response:** We have rephrased this sentence as suggested.

**Figure 2: The calculated H2O signatures seem undetectable in the measured spectrum - is the slant column used for the calculation not matched properly? In the figure description, please specify SZA of the observation and total integration time of the measurement shown.**

**Figure 2:** We now specify the SZA and total integration time in the caption. We also investigated the apparent mismatch between the modeled and measured $H_2O$ signatures, and the slant column has been corrected accordingly.

**Section 3.2: in the opening section of this treatment, information needs to be given concerning which quantities are fitted in the retrieval (please add a table listing all components of the state vector): I guess in addition to the gas mixing ratios of CO2 and H2O, spectral shift (or scale) is fitted? Further fit variables are needed for describing the continuum background level. Is the solar spectral abscissa scale fitted (usually required in spectrally high-res measurements to compensate for residual LOS errors)?**

**Section 3.2, line 126: "the ith measurement" -> "the ith spectral channel in the measured spectrum"**

**Section 3.2:** We have added a table in Section 3.2 listing all components of the state vector used in the retrieval. These include the gas volume mixing ratios of $CO_2$ and $H_2O$. As for the spectral shift and the solar spectral abscissa correction, these are handled during the preprocessing of the spectra. Prior to retrieval, a spectral alignment is applied by calibrating against a stable, unsaturated $H_2O$ absorption line. A scaling factor $\alpha$ is derived from the observed and theoretical line positions to correct the solar spectral abscissa. This correction is performed during preprocessing and is not part of the state vector., but their effects are included via alignment steps prior to the inversion.

**line 126:** This expression has been corrected.

**Section 3.3: unfortunately, the construction of the a-priori (see table 1) is so oversimplified that I doubt any useful conclusions can be drawn from the current information content analysis. If one wants to compare the expected performance of the presented LHR with existing FTIR setups in a sensible manner, the a-priori covariance matrix needs to be far more realistic. Moreover, the information content analysis needs to discuss explicitly the expected errors on column-averaged abundances, which are the products of current networks. Note that the current networks do not provide gas columns, but XGAS values, which are constructed with the help of co-observed O2 columns. Your claim "The LHR exhibits unique advantages … in retrieving gas columns with better vertical discretization [should be: resolution]. It is therefore a promising alternative instrument for local scale measurements or for satellite validation". This might be correct, but needs to be supported by the results of information content analysis. In the application context you refer to in the manuscript (especially satellite validation), high vertical resolution is mainly useful via improving the reconstruction of XGAS amounts over current techniques. (It needs to be kept in mind that the satellite also measures XGAS.)**

**Section 3.3 and Information Content Analysis:** We acknowledge that the $S_a$ matrices used were oversimplified; however, we would like to point out that this was done in the context of comparison with another previous study, which motivated the use of these simplified covariance matrices. We have now addressed this in greater detail and discuss it in the following section.

**For achieving a meaningful comparison with current state-of-the-art, I would suggest to proceed in the following manner:**

**(1) construct sensible S_a matrices for CO2, H2O, and T. Note that the relevant variability here to be reported in S_a is the variability between the actual profile and the TCCON a-priori. This S_a matrix for CO2 can be constructed from aircore launches (the French community is quite active with this technique, so a sufficient amount of data should be available for constructing an S_a matrix). The equivalent matrices for H2O and T can be derived from meteorological soundings, ideally launches which were not used in the model**

assimilation underlying TCCON a-prioris (perhaps H2O and T are by-products of aircore launches anyway?). When constructing the S_a matrices, it is crucially important to maintain the covariances, which inform about characteristic lengths of variability along the vertical. Only by maintaining the diagonal elements a meaningful S_a matrix is constructed.

**(2)  For the performance comparison with TCCON and COCCON for the XGAS values, the propagation of T errors into a O2 retrieval from the 1.26 um band needs to be included. This will alter (expectedly improve) the uncertainty budget for the target quantity XCO2, as this is calculated using the ratio of CO2 and O2 columns. This error compensation is lacking in the LHR approach. Moreover, note that SZA errors cancel out in this rationing approach, so in the discussion of model errors, the resulting error contribution for the LHR needs to be estimated (from the assumed SZA errors).**

**(3) A further important model parameter is the ground pressure. Ideally, it should be included in the error analysis, as the sensitivity of the LHR very likely differs from that of TCCON and COCCON due to the high spectral resolution and due to the fact, that there is no rationing over the O2 column. But if you clearly state in the text that you assume the availability of an ideal sensor, one might skip this item.**

**Using (1) and (2) and your error propagation equations, you can realistically establish the desired performance comparison between the LHR and current techniques. I would expect that the LHR is superior wrt the smoothing error, while the current networks might be more robust wrt the impact of model parameter errors (T and SZA). With respect to the smoothing error, it might be interesting for TCCON + COCCON to work out the smoothing error both for the operational setup (scaling retrieval) and a possible future data processing which performs a profile retrieval fit of CO2. The latter result would specifically reveal the improvement introduced by the high spectral resolution achieved by the LHR. If, however, you feel this is beyond the scope of your work, restrict the investigation to the operational setup.**

(1) Construction of realistic prior covariance matrices ($S_a$):

- We constructed realistic $S_a$ matrices for $CO_2$ and temperature using publicly available AirCore datasets, notably from the MAGIC campaign, covering the period from 2016 to 2023. For $H_2O$, we used data from the ERA5 reanalysis over the same period.

- In all cases, full covariance matrices were retained, including off-diagonal elements, to preserve the vertical correlation lengths. This approach ensures more realistic estimates of the smoothing error and enhances the accuracy of the vertical information content assessment. The diagonal covariance matrices are also kept as a comparison tool.

A short explanation of the method has been added to Section 4.1 of the manuscript.

**(2) XGAS error propagation: temperature and SZA**

We appreciate this important point regarding the propagation of temperature and SZA errors in a column-ratio retrieval framework. However, we respectfully note that in our current LHR configuration, we are not yet able to retrieve $O_2$ columns, as we lack a laser source covering the 1.26 µm $O_2$ absorption band. Procuring such a laser is a planned future upgrade to enable direct XCO2 retrieval via the $CO_2/O_2$ column ratio, consistent with the approach used in TCCON and COCCON. In the absence of an $O_2$ measurement, we do not currently compute $XCO_2$, and the uncertainty budget is expressed in terms of vertically integrated $CO_2$ profile uncertainty, rather than in terms of $XCO_2$. To clarify this and avoid confusion with standard total column quantities, we have revised our terminology throughout the manuscript. Specifically, we now refer to the 'total column uncertainty' as the 'integrated profile uncertainty' to clearly distinguish it from column-averaged quantities like $XCO_2$. We plan to quantify the error cancellation benefits once $O_2$ retrievals become available with the upgraded setup.

Regarding solar zenith angle errors, we agree that these can partially be cancelled when using a gas ratio approach. However, since our current implementation does not include such a ratio, we retain the SZA uncertainty contribution in the LHR error budget for completeness.

**(3) Ground pressure as a model parameter**

We agree that ground pressure is an important parameter, particularly given the high spectral resolution of the LHR and the absence of $O_2$-based column normalization as used in TCCON and COCCON. In our current setup, we use ground pressure measurements from a Vaisala PTU radiosonde with an accuracy of ±0.3 hPa. These values are used to overwrite the default ground pressure in the retrieval algorithm prior to inversion. Given the high accuracy of this input, and to keep the focus on dominant sources of uncertainty, we have not included ground pressure in the error analysis in this study. However, we acknowledge its potential impact and will consider its contribution explicitly in future studies.

**Profile fit:**

We agree that a comparison including the smoothing error from both the operational TCCON/COCCON scaling retrieval and a profile retrieval approach would be highly informative, particularly in demonstrating the benefits of the LHR's high spectral resolution. However, such a detailed comparison would require a significant expansion of the analysis, which we consider beyond the scope of the present work.

Nevertheless, we acknowledge the importance of this direction, and we are currently developing a full profile retrieval fit for $CO_2$ with the LHR. This will allow a more direct assessment of the smoothing error and will be the subject of a dedicated future study. For the current manuscript, we therefore restrict our analysis to the operational scaling retrieval approach.

**In table 3, please add integration times, otherwise the comparison of SNR figures is not meaningful.**

**Table 3:** Integration times have been added for all configurations to allow for meaningful SNR comparison.

**In table 4, the column errors seem unrealistically large to me for all instruments. It therefore would be good to split the reported error into different contributions (spectral noise, model parameters, smoothing). This would make transparent which error source drives the total budget and would allow to explicitly verify at least the calculated noise error contribution, as this can be easily deduced from data retrieved from actual measurements.**

**Table 4:** We now decompose total column errors (integrated profile uncertainty) into:

- Model parameter uncertainty (T, SZA),

- Smoothing error,

- Measurement error.

This helps clarify the dominant sources of uncertainty in each technique.

**The treatment provided in section 4 needs to be refined. The authors claim that from this investigation, the preferred CO2 channels to be measured can be deduced, saving observation and data analysis time. This is in principle correct, but we need to realize that the presented LHR is operated as a ground-based solar absorption spectrometer. In this configuration, a model for the continuum background level needs to be included in the fit (because a solar reference measurement outside of the atmosphere is not doable). This in turn requires a sufficient number of background channels (spectral positions largely free of absorption) to be included both in the measurement and in the fit. The analysis, however, suggests using only channels with strong CO2 absorption, which seems unrealistic.**

**Response:** We have revised this section to emphasize that realistic retrievals require not only strong $CO_2$ absorption channels, but also the identification of a set of informative channels that includes spectral regions free of absorption (i.e., baseline). The new analysis is calculated with a Jacobian that includes this baseline. Furthermore, we now present the top 100 channels ranked by information content, and importantly, we find that nearly 30% of these selected channels are located in baseline regions with little or no $CO_2$ absorption.

---

## Author Response (AR1)

Please find attached a point-by-point response to the reviewers' comments. All relevant changes made to the manuscript have been clearly indicated and are summarized in the response document.

---

## Author Response (AR2)

**Author's response**

This document provides detailed responses to the editor's comments and the main reviewer comments that require substantive revisions. For the reviewer responses, we focus on addressing the major and general comments. Minor corrections and line-by-line editorial suggestions from the reviewers are not included here, as they have already been addressed in our individual responses to the reviewer reports.

**Editor**

**Comment (Editor): Section 3.1, line 90**

*I can hardly believe a Gaussian line shape is assumed for modelling the absorption lines; I guess you refer to the Voigt profile?*

**Author's response:** You are correct, and as the reviewer also pointed out, the Voigt profile is used to model our spectral lines, while the Instrument Line Shape (ILS) is Gaussian. The simulated spectrum is therefore the result of a convolution between the Voigt profile and this Gaussian ILS.

**Change in manuscript:**

Page 4, lines 109–111: The absorption spectrum of gases is derived using the updated HITRAN 2020 database (Gordon et al., 2022), with spectral lines represented by Voigt profiles. The resulting spectrum is convolved with a Gaussian Instrument Line Shape (ILS), which reflects the optical and detection characteristics of the LHR system.

**Comment (Editor): Section 3.1, line 102**

*Please rephrase: "… while TCCON a-priori information is used for $CO_2$ and $H_2O$ atmospheric profiles."*

**Author's response:** We have rephrased this sentence as suggested.

**Change in manuscript:**

Page 5, line 127-129: A priori profiles of CO2 and H2O used to construct the state vector and prior covariance matrix are derived respectively from the AirCore launches from the MAGIC campaigns (see Section 4.1) and the Orléans TCCON station, which is the closest operational site to Dunkirk.

**Comment (Editor): Figure 2**

*The calculated $H_2O$ signatures seem undetectable in the measured spectrum — is the slant column used for the calculation not matched properly? In the figure description, please specify SZA of the observation and total integration time.*

**Author's response:** We now specify the SZA and total integration time in the caption. We also investigated the mismatch between the modeled and measured $H_2O$ features. The slant column has been adjusted accordingly to improve agreement.

**Change in manuscript:**

Page 6, Figure 2 caption: Comparison of measured and simulated LHR transmittance spectra under clear-sky conditions in Dunkirk, for an SZA of 55° and a total integration time of 15 minutes.

**Comment (Editor): Section 3.2**

*In the opening section of this treatment, information needs to be given concerning which quantities are fitted in the retrieval (please add a table listing all components of the state vector)…*

**Author's response:** We have added a table in Section 3.2 listing all retrieval parameters in the state vector, including the volume mixing ratios of $CO_2$ and $H_2O$. Spectral alignment parameters such as shift and scale are handled during preprocessing, not in the state vector. A stable $H_2O$ absorption line is used for spectral calibration.

**Change in manuscript:**

Page 9, Table 1: Table 1 is updated

Page 6, line 148-150: A scaling factor $\alpha$ is derived from the observed and theoretical line positions to correct the solar spectral abscissa. This correction is performed during preprocessing and is not part of the state vector.

**Comment (Editor): Section 3.3 – Prior covariance and information content**

*The construction of the a-priori (see Table 1) is so oversimplified…*

**Author's response:**

- We constructed realistic $S_a$ matrices for $CO_2$ and temperature using publicly available AirCore datasets, notably from the MAGIC campaign, covering the period from 2016 to 2023. For $H_2O$, we used data from the ERA5 reanalysis over the same period.

- In all cases, full covariance matrices were retained, including off-diagonal elements, to preserve the vertical correlation lengths. This approach ensures more realistic estimates of the smoothing error and enhances the accuracy of the vertical information content assessment. The diagonal covariance matrices are also kept as a comparison tool.

A short explanation of the method has been added to Section 4.1 of the manuscript.

**Change in manuscript:**

Page 8-13: We now use realistic prior covariance matrices for $CO_2$ and T from the MAGIC AirCore dataset (2016–2023) and for $H_2O$ from ERA5 reanalysis. These sections are revised and changed.

**Comment (Editor): XGAS error propagation: T and SZA**

*... the propagation of T errors into an $O_2$ retrieval from the 1.26 µm band needs to be included...*

**Author's response:** We appreciate this important point regarding the propagation of temperature and SZA errors in a column-ratio retrieval framework. However, we respectfully note that in our current LHR configuration, we are not yet able to retrieve $O_2$ columns, as we lack a laser source covering the 1.26 µm $O_2$ absorption band. Procuring such a laser is a planned future upgrade to enable direct $XCO_2$ retrieval via the $CO_2/O_2$ column ratio, consistent with the approach used in TCCON and COCCON. In the absence of an $O_2$ measurement, we do not currently compute $XCO_2$, and the uncertainty budget is expressed in terms of vertically integrated $CO_2$ profile uncertainty, rather than in terms of $XCO_2$. To clarify this and avoid confusion with standard total column quantities, we have revised our terminology throughout the manuscript. Specifically, we now refer to the 'total column uncertainty' as the 'integrated profile uncertainty' to clearly distinguish it from column-averaged quantities like $XCO_2$. We plan to quantify the error cancellation benefits once $O_2$ retrievals become available with the upgraded setup.

Regarding solar zenith angle errors, we agree that these can partially be cancelled when using a gas ratio approach. However, since our current implementation does not include such a ratio, we retain the SZA uncertainty contribution in the LHR error budget for completeness.

**Change in manuscript:**

Page 12-13, Section 5.2: To clarify that $XCO_2$ is not retrieved in our current system, we now refer to the reported uncertainty as 'integrated profile uncertainty'. This section is corrected accordingly.

**Comment (Editor): Ground pressure as a model parameter**

*Ideally, it should be included in the error analysis...*

**Author's response:** We agree that ground pressure is an important parameter, particularly given the high spectral resolution of the LHR and the absence of $O_2$-based column normalization as used in TCCON and COCCON. In our current setup, we use ground pressure measurements from a Vaisala PTU radiosonde with an accuracy of ±0.3 hPa. These values are used to overwrite the default ground pressure in the retrieval algorithm prior to inversion. Given the high accuracy of this input, and to keep the focus on dominant sources of uncertainty, we have not included ground pressure in the error analysis in this study. However, we acknowledge its potential impact and will consider its contribution explicitly in future studies.

**Change in manuscript:**

Page 5, lines 124-126: The calculations depend on the concentration of the target atmospheric profile, along with associated data profile such as temperature, pressure, and relative humidity, which are obtained from a nearby PTU300 Vaisala radiosonde, with manufacturer-specified uncertainties of ±0.2°C for temperature, ±0.3 hPa for pressure, and ±1% for relative humidity.

**Comment (Editor): Table 3 – Add integration times**

**Author's response:** Integration times have been added for all cases to allow for meaningful SNR comparison.

**Change in manuscript:**

Page 14, Table 3: A new column labeled "Integration Time" has been added to all rows.

**Comment (Editor): Table 4 – Decompose column errors**

**Author's response:**

- Measurement error,

- Model parameter uncertainty (T, SZA),

- Smoothing error.

**Change in manuscript:**

Page 13, Table 2: We decompose the errors for the LHR in this manner and refer to the previous study in Table 4 for comparison.

**Comment (Editor): Section 4 – On channel selection**

*Realistic retrievals need background channels in addition to absorption peaks…*

**Author's response:** We have revised this section to emphasize that realistic retrievals require not only strong $CO_2$ absorption channels, but also the identification of a set of informative channels that includes spectral regions free of absorption (i.e., baseline). The new analysis is calculated with a Jacobian that includes this baseline. Furthermore, we now present the top 100 channels ranked by information content, and importantly, we find that nearly 30% of these selected channels are located in baseline regions with little or no $CO_2$ absorption.

**Change in manuscript:**

Page 16, lines 361–364: Additionally, in **Figure 5**, we present the first 100 selected channels ranked by their information content with respect to our Jacobian. The first 30 channels are shown in red, channels 31 to 60 in blue, and channels 61 to 100 in green. Notably, the information is primarily concentrated around three absorption lines in the range 6362-6365 cm$^{-1}$. Interestingly, nearly 30% of the top 100 channels lie in baseline regions with little to no $CO_2$ absorption.

**Reviewer 1**

**Comment (Reviewer 1): Acronyms**

*Many acronyms are not explained (TCCON, COCCON, MAGIC, FORUM). With some this might be fine (i.e., citation to the network main paper), but with others not.*

**Author's response:** We have carefully revised the manuscript to ensure that all acronyms are defined at their first occurrence. Where appropriate, we also cite the primary publications describing each network or mission to guide the reader toward more detailed information.

**Change in manuscript:**

Page 2, lines 32 (example): The COllaborative Carbon Column Observing Network (COCCON)...

**Comment (Reviewer 1): State and measurement vectors**

*You write a lot about your model and retrieval, but in the end it is a bit vague what exactly you use in your state vector (for the gases: mixing ratios, concentrations, column densities) and in your measurement vector (radiances, transmittance, ...). I would consider this the most relevant information on a higher level of how your retrieval is designed.*

**Author's response:**

To clarify the retrieval design, we have added a dedicated paragraph at the beginning of Section 3.2 that describes the structure of both the state and measurement vectors:

- The **state vector** contains the vertical profile of $CO_2$ volume mixing ratios (VMRs) on a fixed grid. Depending on the setup, it may also include temperature scaling parameters and interfering species (e.g., $H_2O$).

- The **measurement vector** consists of calibrated radiance spectra derived from solar absorption observations. These are processed using the high-resolution solar reference spectrum from SOLAR-ISS (see also our response to the comment on Line 94).

**Change in manuscript:**

Page 6, lines 144–150: Depending on the retrieval scenario, the state vector may also include additional parameters, such as a scaling factor for atmospheric temperature. The measurement vector $y$ comprises calibrated radiance spectra derived from observed solar absorption, computed by multiplying the solar spectrum (transmittance) with the SOLAR-ISS spectrum (see Section 3.1). Prior to retrieval, all measured spectra are corrected for spectral shift and solar abscissa scale by calibrating against a stable, unsaturated $H_2O$ absorption line. A scaling factor $\alpha$ is derived from the observed and theoretical line positions to correct the solar spectral abscissa. This correction is performed during preprocessing and is not part of the state vector.

**Comment (Reviewer 1): Information Content Analysis – Missing Considerations**

*Regarding your results of the information content analysis of the spectrum (Figure 4 and 5), I am not completely convinced, since I miss a few points in the discussion:*

1. *You do all of this analysis in some type of absorbance space—but to get there from measured radiances, a "background channel" is definitely needed, which I do not see represented in your results.*

2. *You are only considering the $CO_2$ information at the moment, but the large advantage of using a widely tunable laser is that you can measure full rot-vib bands and get constraints on temperature—degenerate with gas amount for a single line or few lines.*

3. *Are you proposing to simply limit the used channels in a retrieval or also to limit the measured spectral bandwidth?*

**Author's response:**

1.      If the "background channel" here refers to a reference radiance spectrum without $CO_2$ absorption (a clean solar background spectrum), unfortunately, such a measurement is not possible in our case because atmospheric absorption is always present along the sunlight's path. Instead, we adopted a commonly used method in solar absorption spectroscopy: we applied a baseline fitting procedure over a broad spectral window to approximate the background continuum. At the same time, we corrected for variations in sunlight and local oscillator laser intensity. This gives us a transmittance spectrum without needing a separate background measurement. Also, while a lab-based LHR system can measure its own heterodyne background, this isn't a valid replacement for the solar background.

2.      We agree that one of the major advantages of using a widely tunable diode laser is its ability to span entire rotational-vibrational bands, providing sensitivity to temperature through the shape and relative intensities of spectral lines. However, the primary aim of this paper is to introduce a new, time-efficient LHR system that can be deployed in field campaigns and achieves accurate retrievals comparable to those from FTIR systems. Unlike FTIR, the larger the spectral range we cover, the longer the acquisition time. Therefore, it's important to find a good balance between spectral coverage and acquisition duration. The integration time should not exceed 5 minutes; otherwise, the air column may become too mixed and the optical path length may vary.

3.      Our intention is to identify and prioritize informative spectral channels within the measured range to improve retrieval stability and reduce computational cost. Depending on the scenario, and specifically in campaign measurements, we propose to limit the measurement bandwidth, especially given the high integration time.

**Change in manuscript:**

Page 16, lines 364–365 (example for 3): This suggests that, in future acquisitions, the combined range can be used to enable faster measurements while preserving a small scan step.

**Reviewer 2**

**Comment (Reviewer 2): Manuscript organization and clarity**

*While the scientific content is of interest, the organization of the manuscript would benefit from improvement… I suggest restructuring Section 3 to reflect a clearer progression: Theory → Application → Results → Comparison. Also, the current use of "a priori" in Sections 3.3.2 and 3.3.3 is confusing.*

**Author's response:** We have reorganized Section 3, as the reviewer asked, as follows:

- Section 3: Theory now includes the forward model and information content analysis (3.1 and 3.2).

- Section 4: Application to the LHR Instrument, now includes the specifics of the a priori state, measurement errors, and non-retrieved parameter treatment (revised from 3.3).

- Section 5: Results → Information Content and Uncertainty, contains the analysis based on LHR simulations (previously 3.4).

- Section 6: Comparison with Existing Networks, presents the comparison with TCCON and COCCON systems (previously 3.5).

We have also clarified the use of the term "a priori" in Section 4.1. In our revised manuscript, we now define this term more precisely to include parameters such as temperature and humidity profiles that are not retrieved but are incorporated as input into the forward model with associated uncertainties. These inputs contribute to the total error budget and are treated using an ensemble of perturbations, as clarified in Section 4.3.

**Change in manuscript:**

- Page 4–14: Sections reorganized to match the logical structure: Theory → Application → Results → Comparison.

- Page 6, lines 140–150: Clarified distinction between retrieved state vector and non-retrieved model inputs.

**Comment (Reviewer 2): Clarification of EM27/SUN and spectral resolution**

*In the introduction (lines 30–36), I recommend expanding the description of the EM27/SUN spectrometer to improve clarity. For example, line 31 should clearly refer to it as the Bruker EM27/SUN, and you can also include the spectral resolution for comparison against the earlier stated IFS125HR spectral resolution...*

**Author's response:**

- In the Introduction (lines 30–36), we now explicitly refer to the Bruker EM27/SUN, and include its nominal spectral resolution of 0.5 cm$^{-1}$, in contrast to the IFS125HR's 0.02 cm$^{-1}$.

- We have revised the sentence about portability and spectral resolution to clarify that reduced resolution arises from design trade-offs in optical path length due to compactness, not portability per se.
- We now cite Herkommer et al. (2024) and Mostafavi Pak et al. (2023) to highlight that the EM27/SUN still performs remarkably well in $CO_2$ retrievals. Please refer to answer 6 to reflect on whether the increased resolution of LHR leads to meaningful improvements in retrieval accuracy.

**Change in manuscript:**

- Page 2, lines 33–40: These spectrometers are relatively easy to operate and enable measurements in locations inaccessible to larger systems, with a spectral resolution of $0.5 \text{ cm}^{-1}$ (Table 3), a trade-off from their compact design which limits the maximum optical path difference. While their portability allows for flexible deployment, maintaining network-wide consistency and coordination remains a significant logistical and technical achievement (Frey et al., 2019). Moreover, several studies have directly compared the performance of the high-resolution IFS125HR with the portable EM27/SUN spectrometers, including Pak et al. (2023) and Herkommer et al. (2024), showing that $CO_2$ retrievals from the EM27/SUN differ by only about 0.1%, a remarkable result considering its lower spectral resolution.

**Comment (Reviewer 2): Radiosonde and ancillary data**

*In Section 3.1, where you describe the use of PTU Vaisala radiosondes and ancillary data from the TCCON database, I suggest adding more specific information to improve transparency and reproducibility…*

**Author's response:** We have expanded the description in Section 3.1 (now Section 3) as follows:

- For the PTU Vaisala radiosonde (PTU300), we now provide typical manufacturer-specified uncertainties: ±0.2°C (temperature), ±0.3 hPa (pressure), and ±1% RH. These values are referenced and used to estimate perturbations in temperature and humidity profiles for the uncertainty analysis in Section 4.3.

- We clarify that ancillary data refers to a priori profiles of $CO_2$ and $H_2O$ used to construct the state vector and prior covariance matrix. In our case, these are derived respectively from the AirCore launches from the MAGIC campaigns and the Orléans TCCON station, which is the closest operational site to Dunkirk from 2016 to 2023.

**Change in manuscript:**

- Page 5, lines 124–129: The calculations depend on the concentration of the target atmospheric profile, along with associated data profile such as temperature, pressure, and relative humidity, which are obtained from a nearby PTU300 Vaisala radiosonde, with manufacturer-specified uncertainties of ±0.2°C for temperature, ±0.3 hPa for

pressure, and ±1% for relative humidity. A priori profiles of $CO_2$ and $H_2O$ used to construct the state vector and prior covariance matrix are derived respectively from the AirCore launches from the MAGIC campaigns (see Section 4.1) and the Orléans TCCON station, which is the closest operational site to Dunkirk.

**Comment (Reviewer 2): Averaging kernel comparison**

*In Section 3.5, include a plot comparing averaging kernels from LHR and FTS instruments.*

**Author's response:** We agree that a direct visual comparison would enhance the interpretation of our results. However, overlaying the averaging kernels significantly reduces the clarity of the figure, as more than 160 lines become indistinguishable. Therefore, we refer the reader to our previous study for a detailed comparison of these averaging kernels.

**Change in manuscript:**

Page 14, lines 310-313: A comparison of averaging kernels (cf. El Kattar, Auriol and Herbin, 2020) with FTS instruments reveals sharper peaks and a more homogeneous vertical distribution than CHRIS, EM27/SUN and IFS125HR, suggesting higher sensitivity at higher altitudes though the a posteriori error $S_x$ is significantly reduced in the lower atmosphere.

**Comment (Reviewer 2): "Channel selection" terminology**

*The term "channel" may be misleading; consider using "line selection" or define your usage clearly.*

**Author's response:** To avoid confusion with terminology used in the TCCON and EM27/SUN communities, we have now explicitly defined the term "channel" at the beginning of Section 7 (previously Section 4). In this study, "channel" refers to an individual spectral point (i.e., a specific wavenumber bin) in the measured radiance spectrum. We have also updated the caption of Figure 5 to reflect this definition and added the term "micro-window selection" where appropriate to clarify that this selection is based on information content per spectral point.

**Change in manuscript:**

- Page 15, line 326-327: To optimize acquisition, we preselect the most informative spectral points, hereafter referred to as channels, prior to measurement. Each channel corresponds to an individual wavenumber bin in the radiance spectrum.

- Page 18, Figure 5 caption updated to reflect this definition.

**Comment (Reviewer 2): 2.74% uncertainty vs. mature networks**

*The reported 2.74% $CO_2$ uncertainty at 10° SZA appears high. TCCON reports 0.16% for $XCO_2$. How does the LHR improve over existing systems?*

**Author's response:** We fully agree that the current level of uncertainty appears high compared to the operational performance of mature networks such as TCCON and COCCON. However, we would like to clarify that the reported 2.74% corresponds to the vertically integrated profile retrieval uncertainty, not to a total column $XCO_2$ uncertainty derived from a ratio of $CO_2$ and $O_2$ columns as in TCCON/COCCON. Since our current setup does not yet include an $O_2$ channel (due to the lack of a suitable laser source in the 1.26 μm region), a true $XCO_2$ product cannot yet be derived. For this reason, and to avoid confusion, we have renamed the reported quantity "integrated profile uncertainty" in the revised manuscript.

We agree that the high spectral resolution of the LHR holds great potential to reduce smoothing errors and improve retrieval quality. A full profile retrieval for $CO_2$ is currently under development and will be presented in a future study. We expect that this, combined with the future addition of an $O_2$ channel, will enable a direct and fair comparison with TCCON/COCCON $XCO_2$ error budgets, including potential advantages in vertical sensitivity. In this study, we focus on the initial demonstration of information content and error propagation for a profile retrieval from a compact LHR instrument, while acknowledging that further development is needed before it can match or surpass operational standards for satellite validation.

**Change in manuscript:**

Page 13, lines 282–287: It's important to note that TCCON's method removes systematic errors common to both the target gas and $O_2$ columns, which is not possible here. In our current LHR configuration, we are not yet able to retrieve $O_2$ columns, as we lack a laser source covering the 1.26 μm $O_2$ absorption band. Procuring such a laser is a planned future upgrade to enable direct $XCO_2$ retrieval via the $CO_2/O_2$ column ratio, consistent with the approach used in TCCON and COCCON. In the absence of an $O_2$ measurement, we do not currently compute $XCO_2$, and the uncertainty budget is expressed in terms of vertically integrated $CO_2$ profile uncertainty, rather than in terms of $XCO_2$.